# Species-specific structural adaptation of the potyviral coat protein in virions and virus-like particles
Neža Koritnik [1,2], Andreja Kežar [1], Luka Kavčič[1], Magda Tušek Žnidarič[3], Adrijana Leonardi[4], Swarnalok De[5,6], Maija Pollari[5], Kristiina Mäkinen [5] ✉ & Marjetka Podobnik [1] ✉

Potyviruses are the largest group of plant positive-sense single-stranded RNA viruses and represent a major economic burden worldwide. Their coat protein (CP) forms a filamentous, flexible capsid around the genomic RNA. However, information is still lacking on the mechanisms of virion assembly, disassembly and stability, which is central to understanding virus biology and control. Here, we investigate the role of CP in these processes using structural, biochemical and biophysical studies of five potyviral CPs from three phylogenetic clades combined with bioinformatics and *in planta* experiments. Our results suggest that, while potyviruses have a conserved virion structure, the amino acids forming the CP-CP and CP-RNA interactions leading to this structure are species-specific. We show that the species-specific CP sequence also determines the architecture of RNA-free virus-like particles (VLPs) and the degree of their structural polymorphism. We identify the residues that determine this specificity at distinct S1-S4 interaction sites. In contrast, a highly conserved charged amino acid triad at the CP-CP interface is essential for the stability of virions and RNA-free VLPs. These results contribute to understanding the molecular mechanism of potyviral virion assembly and highlight the significance of the amino acid sequence of selected CPs in potential biotechnological or biomedical applications.

Potyviruses are among the most widespread and economically important plant pathogens, causing considerable damage to many agricultural, industrial, and ornamental plants worldwide[1]. The genus *Potyvirus* is the largest in the *Potyviridae* family and currently comprises 214 species (https://ictv.global/taxonomy). Potyviruses contain an approximately 10 kb long, positive-sense single-stranded RNA genome that codes for eleven proteins, of which the coat protein (CP) is the only structural protein[1–3]. Many destructive potyvirus diseases affect staple food crops, highlighting their impact on food production[4]. Maize and potato are the world's most cultivated cereal and root/tuber crops, respectively[5]. Maize necrotic lethal disease, caused by mixed infection with a potyvirus and a machlomovirus, as well as potato tuber necrotic ring spot disease, caused by the potato virus Y (PVY; *Potyvirus yituberosi*) NTN strain, are significant examples of geographically widespread and destructive diseases[6]. Aphids acquire potyvirus particles from infected plants within a few minutes and transmit them to

healthy plants during subsequent feeding[7]. Due to this rapid spread, aphidicides do not provide effective solution for controlling the disease[8].

It has been suggested for many RNA plant viruses[9], including potyviruses[10], that rather than a specific origin of assembly, the major determinant for the selective packaging of viral RNAs is the temporal and spatial coupling of replicated RNA and virion assembly. However, the mechanism of assembly and disassembly of potyviruses is still unknown. Since intact particles are required for the virus to infect a new host, it is important to understand the molecular interactions necessary for potyvirus particle assembly and stability. Understanding the factors that make particles unstable can aid in developing control measures.

To date, the structures of three potyviruses - watermelon mosaic virus (WMV, *Potyvirus citrulli*)[11], PVY[12], and turnip mosaic virus (TuMV, *Potyvirus rapae*)[13] have been determined using cryo-electron microscopy (cryo-EM). These structures show the conserved architecture of flexible

[1]Department of Molecular Biology and Nanobiotechnology, National Institute of Chemistry, Ljubljana, Slovenia. [2]PhD Program 'Biomedicine', Faculty of Medicine, University of Ljubljana, Ljubljana, Slovenia. [3]Department of Biotechnology and Systems Biology, National Institute of Biology, Ljubljana, Slovenia. [4]Department of Molecular and Biomedical Sciences, Jožef Stefan Institute, Ljubljana, Slovenia. [5]Department of Agricultural Sciences, Faculty of Agriculture and Forestry, University of Helsinki, Helsinki, Finland. [6]Department of Chemistry and Materials Science, Aalto University, Espoo, Finland. ✉e-mail: kristiina.makinen@helsinki.fi; marjetka.podobnik@ki.si

filamentous virions, in which the CP units are arranged in a left-handed helical symmetry around the genomic RNA with 8.8 CP units per helical turn. Each CP unit consists of the central globular core with the conserved RNA pentanucleotide binding site, and the N- and C-terminal regions with an extended structure[11–13]. The two terminal extensions were labelled N-IDR and C-IDR (IDR: intrinsically disordered regions)[14], as the sequence of amino acids suggests a high degree of structural disorder[14,15]. The N-IDR is exposed on the outer surface of the virion. Its N-terminal half is unstructured (uN-IDR) and has been reported to be involved in non-structural functions of CP, e.g., transmission through aphids[16] or seeds[17], host adaptation and evasion of host resistance responses[15,18], and virus long-distance movement[12,19]. The second part of the N-IDR is structured (sN-IDR) and its elongated fold is supported by its interactions with other CP units on the outer virion surface[11–13]. The N-IDR is crucial for virion assembly and enables virion flexibility[12]. The C-IDR also has an elongated structure that is supported by the helical scaffold of genomic RNA in the lumen of the virion. The C-IDR is essential for efficient viral replication but not for virion assembly[12,14].

The bacterially produced CP of PVY (PVYCP) forms virus-like particles that encapsidate single-stranded RNA (PVYVLP$^{h+RNA}$) with the helical (h) structural parameters highly similar to those of PVY[14]. The formation of virion-like VLP$^{h+RNA}$s has also been shown for another potyvirus, sweet potato feathery mottle virus (SPFMV, *Potyvirus batataplumei*)[20]. However, in contrast to the monomorphic SPFMVVLP$^{h+RNA}$s produced in plants[20], the bacterially produced PVYVLPs were polymorphic, where in addition to PVYVLP$^{h+RNA}$s (8 % of all PVYVLPs), two classes of RNA-free filaments were observed[14]. The larger class (67 % of all PVYVLPs) consisted of RNA-free flexible filaments composed of stacked octameric PVYCP rings (r) along the filament axis (PVYVLP$^{r8}$). The smaller class of RNA-free VLPs (25 % of all PVYVLPs), was defined by left-handed helical symmetry with 7.5 CP subunits per turn (PVYVLP$^{h7.5}$). In RNA-free VLPs, the C-IDR was disordered, while the sN-IDR retained the role of the structurally plastic CP-CP linker, facilitating structural polymorphism of PVYVLPs[14].

Due to evolutionary, biological and ecological factors, most potyviruses are specialized to a relatively narrow host range[21–24]. Their genomes are subject to negative selection, with the region encoding CP being one of the most strongly selected[2]. The amino acid identity of CP between potyvirus species is 55–75%[25] (Supplementary Fig. 1a). The greatest variability, both in length and amino acid sequence, is found in the uN-IDR (Supplementary Fig. 1), whose high tolerance to mutations probably facilitates viral adaptation[15]. The amino acid variability of the structured part of the potyviral CP is significantly lower (Supplementary Fig. 1), consistent with the conserved virion architecture[11–13].

Here, we determined the cryo-EM structures of potato virus A (PVA; *Potyvirus atuberosi*) and PVAVLPs. As we observed unique structural features in both the virion and VLPs compared to the corresponding structures formed by PVYCP[12,14], we investigated whether the amino acid sequences of potyviral CPs contain species-specific structural determinants. To address this we performed structural, biochemical and biophysical studies of five potyviral CPs from three phylogenetic clades combined with bioinformatics and *in planta* experiments. We show that, although potyviral virions have a conserved overall quaternary structure, the fold of three regions of CP units can vary between potyvirus species. This variation is due to species-specific amino acid sequences in these regions, which lead to species-specific CP-CP and CP-RNA interaction networks in virions. We also show that the architecture of RNA-free VLPs and the degree of their structural polymorphism depend on the species-specific CP sequence at sites S1 to S4, mainly in the globular CP core, affecting CP-CP interactions in VLPs. The chemical nature of the involved residues appears important, as mutations significantly alter the structural parameters of RNA-free VLPs compared to those produced by wild-type CPs. In contrast to the demonstrated species-specific structural features in virions and VLPs, we found that the stability of these structures and the equilibrium between them depend on a highly conserved charged amino acid triad at the CP-CP interface, contributed by sN-IDR and the globular CP core region. Thus, the balance between the

species-specific and the highly conserved residues at the CP-CP and CP-RNA interfaces, combined with the structural restraints, may represent a regulatory mechanism for the assembly of structurally uniform, stable and functional virions via highly ordered RNA-free intermediates.

## Results

### The cryo-EM structure of potato virus A shows an overall conserved structure with unique details

Potyvirus potato virus A (PVA) was purified from infected *Nicotiana benthamiana* (Supplementary Fig. 2a-c). Its cryo-EM structure was determined with an overall resolution of 2.4 Å (Fig. 1a, b, Table 1), which to our knowledge represents the highest resolution of potyvirus to date (Supplementary Fig. 2d).

The structure of PVA is similar to known structures of potyviruses[11–13], as is the fold of PVACP (Fig. 1c). All residues of PVACP were traced in cryo-EM density, except for the uN-IDR residues Ala1 to Thr43 (Fig. 1b, Supplementary Fig. 2e). The extended structure of the sN-IDR (His44-Lys78) continues into the globular core region (Gln79-Leu227), followed by the structurally extended C-IDR (Lys228-Val269). In PVA, the C-IDR is fully defined, as it is in PVY[12] but not in WMV[11] and TuMV[13] (Supplementary Fig. 3a), likely due to the quality of the cryo-EM maps. We therefore performed comparative analysis between PVACP and PVYCP, which revealed differences in (i) the conformation of N-IDR (PVACP numbering) between residues His44-His71, named as a 'sN-IDR kink', (ii) at a junction between the core α8-helix and the C-IDR (Lys228-Asn231), and (iii) in the C-IDR between Val241-Glu245 (Fig. 1c, Supplementary Fig. 3b, c). These structural differences in regions (i)-(iii) between PVACP and PVYCP align with the markedly low degree of amino acid conservation in these three regions within CPs in the genus *Potyvirus* (Fig. 1d).

In potyviruses, the CP-CP interaction network is mainly based on hydrogen bonds and salt bridges[11–13], (Supplementary Fig. 4a) and sN-IDR greatly contributes to these interactions. In the upper panel of Fig. 1e, we compare the interactions in PVA with those in PVY[12]. CP subunits are numbered as n, n + 1, n + 2, up to n + 17, with corresponding mirror-image positions designated as n-1, n-2, through n-17. For example, in PVY, if CP$^n$ interacts with CP$^{n-2}$, then CP$^{n+2}$ also interacts with CP$^n$ (if one imagines the IDRs drawn for all subunits, as described in detail in our earlier study[12]). Therefore, in contrast to PVY, where each PVYCP unit is in contact with twelve neighboring PVYCPs (PDB ID: 6HXX)[12], the number of neighboring subunits in PVA is ten (Fig. 1e). Namely, the salt bridges in PVY between CP$^n$ Lys58 from sN-IDR kink and CP$^{n-2}$ Asp138 from the β-hairpin, and between CP$^{n+2}$ Lys58 and CP$^n$ Asp138, are absent in PVA, as Met60 and Glu140 are located at the corresponding sites in PVACP (Fig. 1e).

Structural differences between PVA and PVY in regions (ii) and (iii) affect the CP-RNA interactions. In the potyviral CP, the canonical RNA-binding site binds a pentanucleotide, modeled as uracils U1-U5[11–13]. The RNA-binding site is located on the luminal side of the filament, between the core residues Ser127-Gly132 and the region (ii) at the junction of the C-terminal end of the α8-helix and the C-IDR of the same CP[11–13]. In PVACP, the RNA-binding residues in this pocket are Ser127, Arg159, and Asp203, which are conserved in the families *Potyviridae*, *Alphaflexiviridae*, *Betaflexiviridae*, and *Closteroviridae*[11], as well as Gln160, Arg163, Arg185, Tyr186, Arg190 and Ser230 (Fig. 1f). Most of them are conserved in PVYCP, except for Ala161 and Ala228, which correspond to Arg163 and Ser230 in PVACP (Supplementary Fig. 3c), resulting in a reduced CP-RNA network in PVY compared to PVA. The region (iii) is at the second, 'non-canonical' RNA-binding site, where Thr242 in the C-IDR of PVACP$^n$ forms a hydrogen bond with the nucleotide U2 bound in the canonical RNA-binding pocket of PVACP$^{n+9}$ (Fig. 1f), and in PVY, Ser240 plays a similar role[12]. The structural differences in region (iii) could be contributed by Gly238 in PVYCP corresponding to Asn240 in PVACP (Supplementary Fig. 3c).

Thus, compared to PVY, PVA has a smaller CP-CP interaction network and a larger CP-RNA network. This seems to compensate for the differences between the amino acid sequences of the two species, while maintaining the same architecture and comparable thermal stability of the

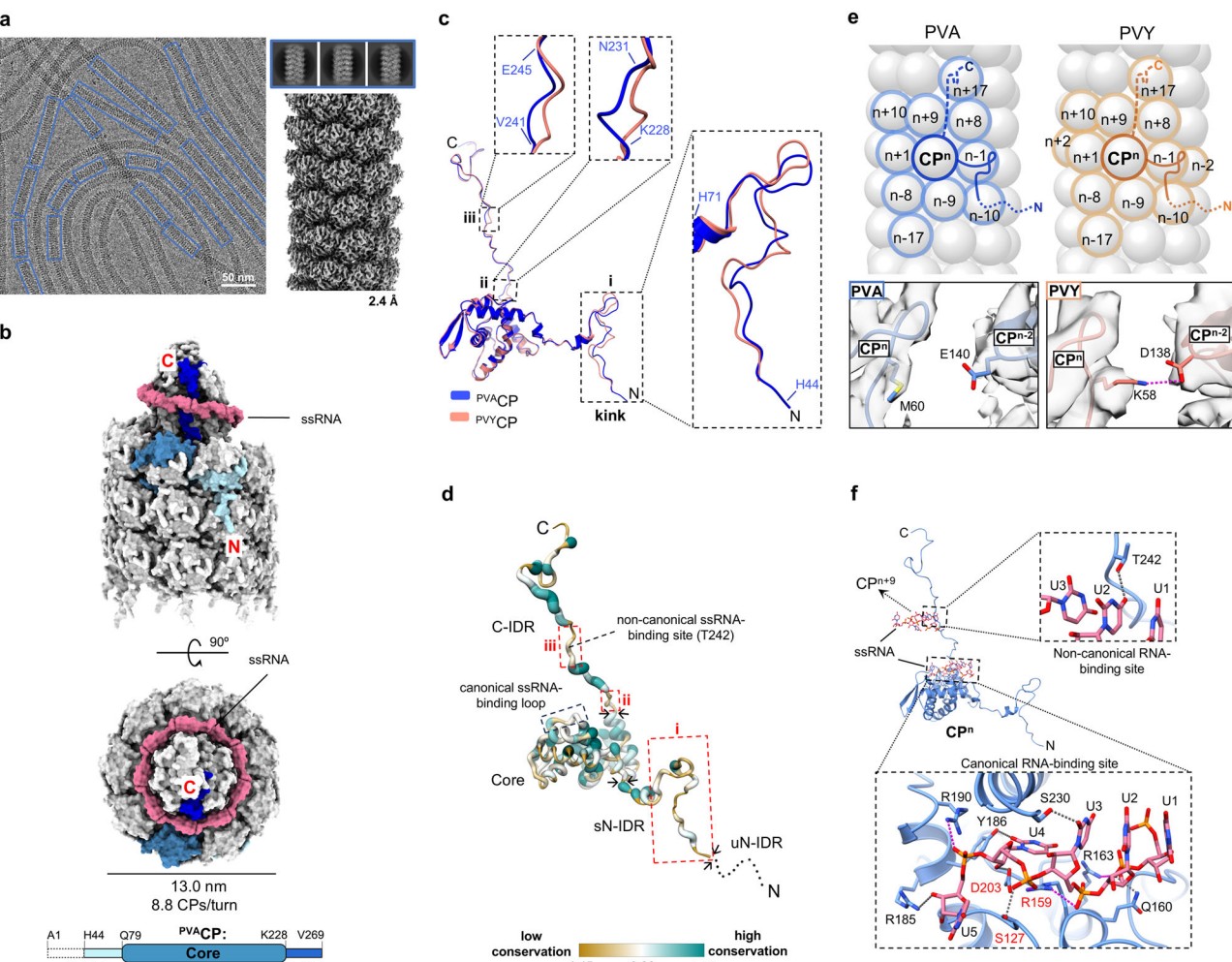

**Fig. 1 | Structural analysis of PVA. a** Cryo-EM micrograph (left), 2D class averages (top right), and cryo-EM density map (bottom right) with the overall resolution in Å. **b** Surface representation of PVA, the side-view (above) and the top-view (below). One $^{PVA}$CP unit is colored in shades of blue, as in the schematic representation (created with BioRender.com) of a $^{PVA}$CP polypeptide chain below; other $^{PVA}$CP units are in light grey. The viral RNA is in pink. The uN-IDR (A1-T43) is shown as a dotted rectangle on a scheme. **c** Structural superposition (all Cα atoms) of $^{PVA}$CP and $^{PVY}$CP (PDB ID: 6HXX). Regions with structural differences are enlarged (dashed frames). **d** Conservation level of amino acids in potyviral CPs mapped onto the $^{PVA}$CP structure. Black arrows indicate the boundaries between the CP core region and sN-IDR or C-IDR and between uN-IDR and sN-IDR. Red dashed rectangles highlight the IDR regions (i-iii) with low sequence conservation. The conservation score was calculated using the AL2CO algorithm[72] in ChimeraX v1.9[58] using a

MUSCLE[66] sequence alignment of 158 potyviral CP sequences from phylogenetic tree currently available at (https://ictv.global/system/files/inline-images/OPSR. Pot_.Fig3_.v15.png). GenBank accession numbers of CP sequences are provided in the Supplementary Data 1 file. **e** Schematic representation of the CP-CP interaction networks in PVA (left) and PVY (right). The focal $CP^n$ unit (dark color) is surrounded by interacting CP units (light color). Bellow, interfaces between $CP^n$ and $CP^{n-2}$ are shown with atomic models of PVA (left, blue) and PVY (PDB ID: 6HXX, EMD-0297; right, orange) in their respective cryo-EM density maps (gray). **f** Interactions between $^{PVA}$CP (blue ribbon) and RNA pentanucleotides (U1-U5, pink sticks). Enlarged views of RNA-CP interaction sites are shown in dashed frames. $^{PVA}$CP residues interacting with RNA are presented in sticks. In (**e**, **f**), the heteroatoms are colored blue (N), red (O), yellow (S) and orange (P). Hydrogen bonds and salt-bridges are shown as dotted lines in dark grey and magenta, respectively.

virions (Supplementary Fig. 4b)[14]. The large network of CP-CP and CP-RNA interactions probably also allows potyviruses to survive *ex planta*. Indeed, we kept the PVA sample at 4 °C for six months, and the cryo-EM structure of the incubated virions, determined at 2.5 Å resolution, remained unchanged (Table 1, Supplementary Fig. 4c–e). However, over time, we observed proteolysis of the surface-exposed uN-IDR up to Gly42 (Supplementary Fig. 4f, g), likely due to protease contamination.

In summary, PVA shares a conserved overall structure with other potyviruses. Compared to PVY, we found structural differences in three regions of CP, which exhibit high degree of amino acid variability among potyviral CPs. One is located at the CP-CP interface, and the other two at the CP-RNA interface. The differences in amino acid sequence between $^{PVA}$CP and $^{PVY}$CP in these regions lead to distinct CP-CP and CP-RNA interaction networks in PVA and PVY, the net sum of which, however, ensures the structural integrity of the conserved capsid.

## $^{PVA}$CP and $^{PVY}$CP form architecturally distinct RNA-free VLPs

Next, we investigated whether the unique sequence of $^{PVA}$CP leads to the formation of structurally unique VLPs. We prepared recombinant $^{PVA}$VLPs (using the same CP sequence as in PVA, i.e., isolate B11) according to the protocol for $^{PVY}$VLPs[14] and analyzed them with cryo-EM. In contrast to $^{PVY}$VLPs[14], we repeatedly observed only two architectural types of $^{PVA-B11}$VLPs (Fig. 2a–c). $^{PVA-B11}$VLP$^{h+RNA}$s have helical parameters comparable to potyviral capsids and other known RNA-packaging VLPs[11–14,20] (Table 1, Supplementary Table 1). The fold of the sN-IDR in $^{PVA-B11}$CP$^{h+RNA}$ matches $^{PVA}$CP better than $^{PVY}$CP, but a unique fold was observed between residues Leu49-Thr57 of $^{PVA-B11}$CP$^{h+RNA}$, demonstrating the structural plasticity of the sN-IDR even within the same amino acid sequence (Fig. 2d).

$^{PVA-B11}$CP formed only one type of RNA-free VLPs. They exhibited left-handed helical symmetry, but unlike $^{PVY}$VLP$^{h7.5}$ (PDB ID: 8OPB)[14], $^{PVA-B11}$CP formed filaments with 8.5 CP units per helical turn ($^{PVA-B11}$VLP$^{h8.5}$), with a

## Table 1 | Cryo-EM data collection, refinement and validation statistics

| | PVA | PVA insolation | PVA-BT VLP | | PVA-Datura VLP | | PVA VLPmut6 | | | PVA VLP D138C | TEV VLP | PopMoV VLP | | JGMV VLP | | | |
|---|---|---|---|---|---|---|---|---|---|---|---|---|---|---|---|---|---|
| Filament architecture | h+RNA | h+RNA | h+RNA | h8.5 | h+RNA | h8.5 | h9.7 | h10.7 | r10 | h+RNA | h8.5 | h+RNA | r9 | h+RNA | h8.6 | h9.6 | r9 |
| EMDB ID | EMD-53790 | EMD-53791 | EMD-53792 | EMD-53793 | EMD-53794 | EMD-53796 | EMD-53799 | EMD-53800 | EMD-53801 | EMD-53802 | EMD-53862 | EMD-53863 | EMD-53864 | EMD-53865 | EMD-53866 | EMD-53867 | EMD-53868 |
| PDB ID | 9R7R | 9R7S | 9R7T | 9R7U | 9R7V | 9R7X | 9R7Y | 9R7Z | 9R80 | 9R81 | 9R9W | 9R9X | 9R9Y | 9R9Z | 9RA0 | 9RA1 | 9RA2 |
| **Percentage of particles detected in the sample (%)** | | | | | | | | | | | | | | | | | |
| Isolation 1 | 100 | 100 | 61 | 39 | 6 | 94 | 64 | 21 | 15 | 100 | 100 | 84 | 16 | 20 | 9 | 16 | 55 |
| Isolation 2 | / | / | 66 | 34 | 49 | 51 | 49 | 29 | 22 | / | 100 | 93 | 7 | 8 | 7 | 20 | 65 |
| Isolation 3 | / | / | / | / | 30 | 70 | / | / | / | / | / | / | / | / | / | / | / |
| **Data collection and processing** | | | | | | | | | | | | | | | | | |
| EMPIAR ID | EMPIAR-12819 | EMPIAR-12820 | EMPIAR-12821 | | EMPIAR-12822 | | EMPIAR-12823 | | | EMPIAR-12824 | EMPIAR-12825 | EMPIAR-12826 | | EMPIAR-12827 | | | |
| Magnification | ×150,000 | ×150,000 | ×150,000 | ×150,000 | ×150,000 | ×150,000 | ×150,000 | ×150,000 | ×150,000 | ×150,000 | ×150,000 | ×150,000 | ×150,000 | ×150,000 | ×150,000 | ×150,000 | ×150,000 |
| Voltage (kV) | 200 | 200 | 200 | 200 | 200 | 200 | 200 | 200 | 200 | 200 | 200 | 200 | 200 | 200 | 200 | 200 | 200 |
| Electron exposure (e–/Å²) | 40 | 40 | 40 | 40 | 40 | 40 | 40 | 40 | 40 | 40 | 40 | 40 | 40 | 40 | 40 | 40 | 40 |
| Defocus range (μm) | -2.0 to -0.8 | -2.0 to -0.8 | -2.0 to -0.8 | -2.0 to -0.8 | -2.0 to -0.8 | -2.0 to -0.8 | -2.0 to -0.8 | -2.0 to -0.8 | -2.0 to -0.8 | -2.0 to -0.8 | -2.0 to -0.8 | -2.0 to -0.8 | -2.0 to -0.8 | -2.0 to -0.8 | -2.0 to -0.8 | -2.0 to -0.8 | -2.0 to -0.8 |
| Pixel size (Å) | 0.95 | 0.95 | 0.95 | 0.95 | 0.95 | 0.95 | 0.95 | 0.95 | 0.95 | 0.95 | 0.95 | 0.95 | 0.95 | 0.95 | 0.95 | 0.95 | 0.95 |
| Symmetry imposed | C1, helical | C1, helical | C1, helical | C1, helical | C1, helical | C1, helical | C1, helical | C1, helical | C10, helical | C1, helical | C1, helical | C1, helical | C9, helical | C1, helical | C1, helical | C1, helical | C9, helical |
| Helical twist (°) | -40.90 | -40.90 | -40.95 | -42.45 | -40.95 | -42.55 | -37.08 | -33.75 | 19.64 | -40.95 | -42.45 | -40.99 | 27.35 | -40.95 | -41.80 | -37.65 | 14.89 |
| Helical rise (Å) | 3.89 | 3.89 | 3.96 | 4.74 | 3.91 | 4.71 | 3.85 | 3.46 | 36.33 | 3.96 | 4.74 | 3.93 | 33.50 | 4.03 | 4.85 | 4.50 | 42.79 |
| Initial segment images (no.) | 566,164 | 701,092 | 930,558 | 930,558 | 147,182 | 147,182 | 519,832 | 519,832 | 519,832 | 557,312 | 335,220 | 543,032 | 543,032 | 748,494 | 748,494 | 748,494 | 748,494 |
| Final segment images (no.) | 481,088 | 629,681 | 291,040 | 185,072 | 6,503 | 96,670 | 122,826 | 71,171 | 49,408 | 319,228 | 113,702 | 36,808 | 2,683 | 31,352 | 30,172 | 81,832 | 271,709 |
| Map resolution (Å) | 2.4 | 2.5 | 2.5 | 3.3 | 3.1 | 2.6 | 3.9 | 3.9 | 3.7 | 2.3 | 3.3 | 3.6 | 4.3 | 3.4 | 4.4 | 4.0 | 3.5 |
| FSC threshold | 0.143 | 0.143 | 0.143 | 0.143 | 0.143 | 0.143 | 0.143 | 0.143 | 0.143 | 0.143 | 0.143 | 0.143 | 0.143 | 0.143 | 0.143 | 0.143 | 0.143 |
| **Refinement** | | | | | | | | | | | | | | | | | |
| Model resolution (Å) | 2.4 | 2.6 | 2.6 | 3.4 | 3.1 | 2.7 | 4.0 | 4.0 | 3.8 | 2.4 | 3.4 | 3.8 | 4.4 | 3.4 | 4.6 | 4.2 | 3.7 |
| FSC threshold | 0.5 | 0.5 | 0.5 | 0.5 | 0.5 | 0.5 | 0.5 | 0.5 | 0.5 | 0.5 | 0.5 | 0.5 | 0.5 | 0.5 | 0.5 | 0.5 | 0.5 |
| **Model composition** | | | | | | | | | | | | | | | | | |
| Protein chains | 36 | 36 | 36 | 24 | 36 | 24 | 27 | 30 | 30 | 36 | 24 | 36 | 27 | 36 | 24 | 27 | 27 |
| RNA chains | 36 | 36 | 36 | 0 | 36 | 0 | 0 | 0 | 0 | 36 | 0 | 36 | 0 | 36 | 0 | 0 | 0 |
| Protein residues | 8,136 | 8,136 | 8,136 | 4,392 | 8,136 | 4,392 | 4,995 | 5,550 | 5,580 | 8,136 | 4,704 | 8,100 | 4,995 | 8,064 | 4,272 | 4,806 | 4,833 |
| Non-hydrogen atoms | 68,940 | 68,940 | 68,940 | 35,592 | 68,940 | 35,592 | 40,419 | 44,910 | 45,120 | 68,868 | 37,968 | 68,508 | 40,527 | 68,976 | 35,376 | 39,798 | 39,987 |
| Ligands | 0 | 0 | 0 | 0 | 0 | 0 | 0 | 0 | 0 | 0 | 0 | 0 | 0 | 0 | 0 | 0 | 0 |
| Protein residues per CP | 269 | 269 | 269 | 269 | 269 | 269 | 269 | 269 | 269 | 269 | 263 | 273 | 273 | 303 | 303 | 303 | 303 |
| Missing N-terminal CP residues | 43 | 43 | 43 | 43 | 43 | 43 | 43 | 43 | 42 | 43 | 37 | 47 | 47 | 77 | 78 | 78 | 77 |
| Missing C-terminal CP residues | 0 | 0 | 0 | 43 | 0 | 43 | 41 | 41 | 41 | 0 | 30 | 1 | 41 | 2 | 47 | 47 | 47 |
| Filament diameter (nm) | 13.0 | 13.0 | 13.0 | 12.5 | 13.0 | 12.5 | 13.5 | 14.5 | 14.0 | 13.0 | 12.5 | 13.0 | 13.0 | 13.0 | 12.5 | 13.5 | 13.0 |
| **R.m.s. deviations** | | | | | | | | | | | | | | | | | |
| Bond lengths (Å) | 0.008 | 0.007 | 0.007 | 0.006 | 0.008 | 0.007 | 0.010 | 0.010 | 0.010 | 0.007 | 0.009 | 0.007 | 0.010 | 0.009 | 0.007 | 0.008 | 0.009 |
| Bond angles (°) | 0.944 | 0.860 | 0.883 | 0.835 | 1.012 | 0.798 | 1.147 | 1.168 | 1.047 | 0.895 | 0.949 | 0.860 | 1.346 | 0.895 | 0.906 | 0.959 | 0.963 |
| **Validation** | | | | | | | | | | | | | | | | | |
| MolProbity score | 1.49 | 1.30 | 1.36 | 1.54 | 1.87 | 1.48 | 1.89 | 1.78 | 1.89 | 1.24 | 1.75 | 1.73 | 2.21 | 1.72 | 2.13 | 1.77 | 1.71 |
| Clashscore | 7.98 | 5.53 | 4.55 | 7.25 | 13.22 | 3.85 | 11.72 | 8.01 | 9.28 | 4.67 | 11.19 | 9.30 | 19.26 | 11.72 | 15.32 | 8.35 | 6.53 |
| Poor rotamers (%) | 0.00 | 0.00 | 0.00 | 0.00 | 0.00 | 0.00 | 0.00 | 0.00 | 0.00 | 0.50 | 0.00 | 0.00 | 0.00 | 0.00 | 0.00 | 0.00 | 0.00 |
| Ramachandran plot | | | | | | | | | | | | | | | | | |

## Table 1 (continued) | Cryo-EM data collection, refinement and validation statistics

| | PVA / PVA^VLP mut5 | PVA^incubation | PVA-BT'VLP | PVA-Datura VLP / PVA^VLP R180A | PVA^VLP mut6 / PVA^VLP Q87H | PVA^VLP mut5 / PVA^VLP Q87H | PVA^VLP R180A / PVA^VLP R180T | PVA^VLP Q87H | PVA^VLP R180T / PVA^VLP D138C | PVA^VLP D138C | TBV VLP / PVA^VLP Q87H-R180T | PepMoV VLP | JGMV VLP / PVA^VLP Q87H-R180T-R180T | PVA^VLP Q87H-R180T | PVA^VLP R180T / PVA^VLP 161R | PVA^VLP 161R | PVA^VLP in vitro |
|---|---|---|---|---|---|---|---|---|---|---|---|---|---|---|---|---|---|
| Favored (%) | 97.77 | 98.66 | 97.32 | 97.24 | 96.43 | 95.58 | 95.61 | 95.03 | 94.02 | 98.49 | 96.91 | 96.41 | 93.44 | 97.30 | 93.11 | 95.45 | 94.92 |
| Allowed (%) | 2.23 | 1.34 | 2.68 | 2.76 | 3.57 | 4.42 | 4.39 | 4.97 | 5.98 | 1.51 | 3.09 | 3.59 | 6.56 | 2.70 | 6.89 | 4.55 | 5.08 |
| Disallowed (%) | 0.00 | 0.00 | 0.00 | 0.00 | 0.00 | 0.00 | 0.00 | 0.00 | 0.00 | 0.00 | 0.00 | 0.00 | 0.00 | 0.00 | 0.00 | 0.00 | 0.00 |
| Correlation coefficient (CC) | 0.83 | 0.85 | 0.86 | 0.82 | 0.86 | 0.85 | 0.83 | 0.85 | 0.85 | 0.85 | 0.85 | 0.85 | 0.80 | 0.87 | 0.85 | 0.84 | 0.86 |
| Filament architecture | h+RNA | h8.7 | h9.7 | r10 | h8.5 | r8 | h+RNA | h8.5 | r8 | h+RNA | h7.5 | h8.5 | r8 | h+RNA | h8.5 | h+RNA | h8.5 |
| EMDB ID | EMD-53632 | EMD-53633 | EMD-53634 | EMD-53635 | EMD-53636 | EMD-53638 | EMD-53639 | EMD-53640 | EMD-53642 | EMD-53643 | EMD-53644 | EMD-53645 | EMD-53646 | EMD-53647 | EMD-53650 | EMD-53651 | |
| PDB ID | / | / | / | / | / | / | / | / | / | / | / | / | / | / | / | / | / |
| **Percentage of particles detected in the sample (%)** | | | | | | | | | | | | | | | | | |
| Isolation 1 | 30 | 41 | 27 | 2 | 100 | 73 | 25 | 2 | 11 | 6 | 32 | 51 | 23 | 77 | 100 | | |
| Isolation 2 | 24 | 16 | 59 | 1 | / | / | / | / | / | / | / | / | / | / | / | | |
| **Data collection and processing** | | | | | | | | | | | | | | | | | |
| EMPIAR ID | EMPIAR-12828 | | | EMPIAR-12829 | EMPIAR-12830 | EMPIAR-12831 | | | | | EMPIAR-12832 | | | EMPIAR-12833 | | EMPIAR-12834 | |
| Magnification | ×150,000 | ×150,000 | ×150,000 | ×150,000 | ×150,000 | ×150,000 | ×150,000 | ×150,000 | ×150,000 | ×150,000 | ×150,000 | ×150,000 | ×150,000 | ×150,000 | ×150,000 | ×150,000 | |
| Voltage (kV) | 200 | 200 | 200 | 200 | 200 | 200 | 200 | 200 | 200 | 200 | 200 | 200 | 200 | 200 | 200 | 200 | |
| Electron exposure (e⁻/Å²) | 40 | 40 | 40 | 40 | 40 | 40 | 40 | 40 | 40 | 40 | 40 | 40 | 40 | 40 | 40 | 40 | |
| Defocus range (μm) | -2.0 to -0.8 | -2.0 to -0.8 | -2.0 to -0.8 | -2.0 to -0.8 | -2.0 to -0.8 | -2.0 to -0.8 | -2.0 to -0.8 | -2.0 to -0.8 | -2.0 to -0.8 | -2.0 to -0.8 | -2.0 to -0.8 | -2.0 to -0.8 | -2.0 to -0.8 | -2.0 to -0.8 | -2.0 to -0.8 | -2.0 to -0.8 | |
| Pixel size (Å) | 0.95 | 0.95 | 0.95 | 0.95 | 0.95 | 0.95 | 0.95 | 0.95 | 0.95 | 0.95 | 0.95 | 0.95 | 0.95 | 0.95 | 0.95 | 0.95 | |
| Symmetry imposed | C1, helical | C1, helical | C1, helical | C10, helical | C1, helical | C8, helical | C1, helical | C8, helical | C1, helical | C1, helical | C1, helical | C8, helical | C1, helical | C1, helical | C1, helical | C1, helical | |
| Helical twist (°) | -40.95 | -41.55 | -37.10 | 19.74 | -42.55 | 14.50 | -40.95 | 14.36 | -41.13 | -48.10 | -42.49 | 14.36 | -41.04 | -42.30 | -40.95 | -42.50 | |
| Helical rise (Å) | 3.97 | 4.53 | 3.85 | 36.33 | 4.73 | 43.54 | 3.92 | 43.54 | 4.09 | 5.88 | 5.18 | 43.50 | 4.05 | 5.37 | 3.94 | 4.73 | |
| Initial segment images (no.) | 539,979 | 539,979 | 539,979 | 539,979 | 924,941 | 454,469 | 454,469 | 454,469 | 138,076 | 138,076 | 138,076 | 138,076 | 402,041 | 402,041 | 298,823 | 871,012 | |
| Final segment images (no.) | 76,521 | 106,287 | 68,748 | 4,863 | 561,447 | 150,486 | 51,600 | 4,707 | 3,398 | 2,030 | 10,439 | 16,689 | 28,564 | 93,829 | 108,921 | 93,491 | |
| Map resolution (Å) | 3.5 | 3.8 | 3.8 | 3.9 | 3.4 | 4.0 | 2.8 | 4.9 | 3.2 | 6.8 | 4.7 | 3.9 | 2.5 | 4.0 | 2.9 | 3.6 | |
| FSC threshold | 0.143 | 0.143 | 0.143 | 0.143 | 0.143 | 0.143 | 0.143 | 0.143 | 0.143 | 0.143 | 0.143 | 0.143 | 0.143 | 0.143 | 0.143 | 0.143 | 0.143 |

Statistics for cryo-EM density maps and structural models is included.
Statistics for cryo-EM density maps.

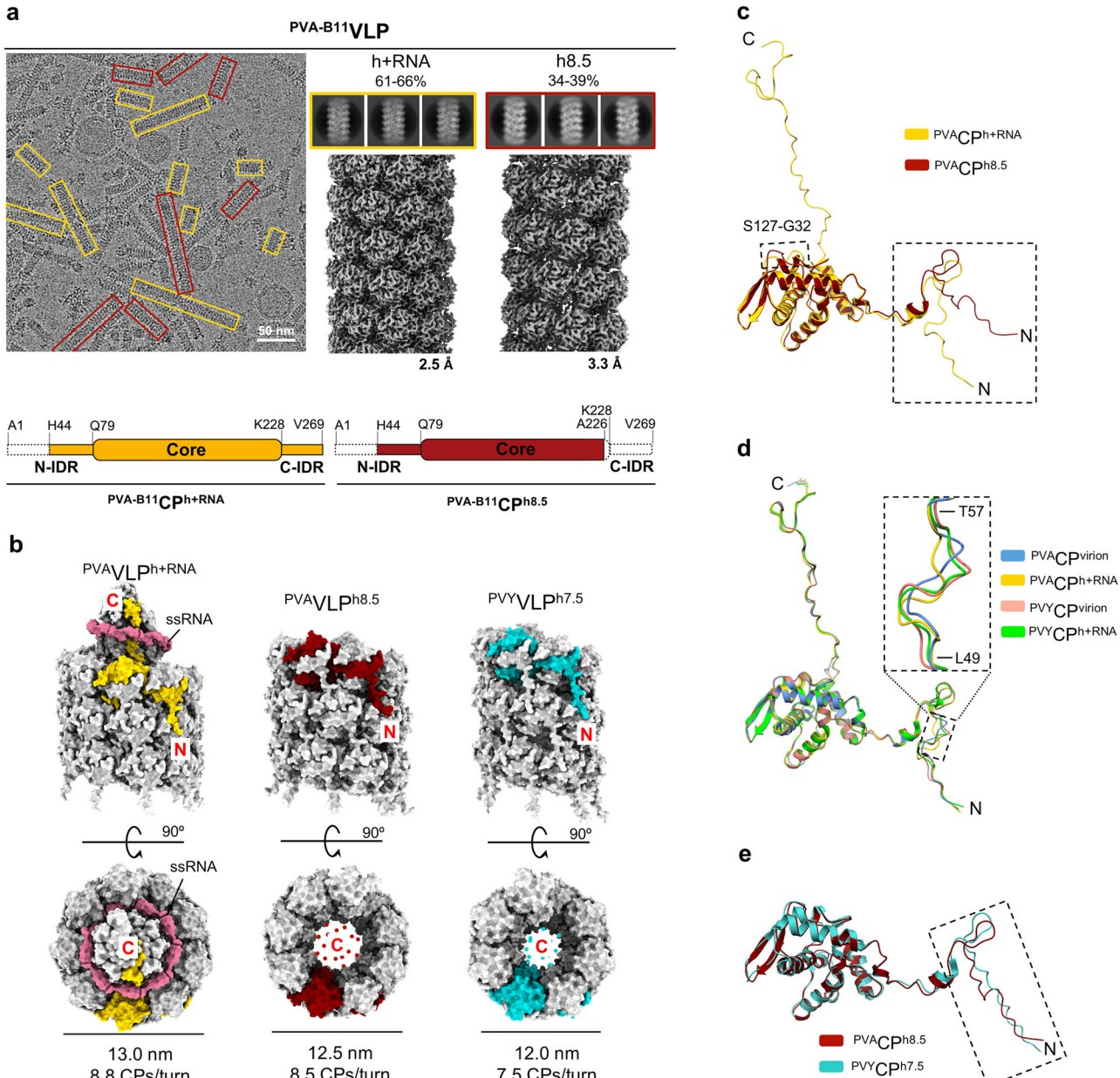

**Fig. 2 | Structural analysis of $^{PVA}$VLPs. a** Top: cryo-EM micrograph (left) of $^{PVA-B11}$VLPs; top-right: 2D class averages of VLP$^{h+RNA}$ and VLP$^{h8.5}$. The % above 2D classes mark proportion of the VLPs of a certain architecture among all VLPs produced by $^{PVA-B11}$CP. Middle-right: cryo-EM density maps with their overall resolution in Å. Architecturally distinct filaments are highlighted in yellow (VLP$^{h+RNA}$) and dark red (VLP$^{h8.5}$). Bottom: schematic representation of $^{PVA-B11}$CP$^{h+RNA}$ and $^{PVA-B11}$CP$^{h8.5}$ polypeptide chains (created with BioRender.com), the dashed lines represent unstructured residues. **b** Surface representation of $^{PVA-B11}$VLPs (VLP$^{h+RNA}$, left; VLP$^{h8.5}$, middle) and $^{PVY}$VLP$^{h7.5}$ (PDB ID: 8OPB, right), viewed from the side (top) and the top (bottom). One CP unit is colored in yellow ($^{PVA}$VLP$^{h+RNA}$), dark red ($^{PVA}$VLP$^{h8.5}$) or turquoise ($^{PVY}$VLP$^{h7.5}$), other CP units are in light grey. RNA is in pink. Dark red or turquoise dots within the VLP$^h$ lumens represent the unstructured C-IDR. Diameters of particles are provided (in nm) and the number of CP units per helical turn. **c** Structural superposition on all Cα atoms of $^{PVA-B11}$CP$^{h+RNA}$ and $^{PVA-B11}$CP$^{h8.5}$. **d** Structural superposition on all Cα atoms of CP units from RNA-packaging particles: PVA, $^{PVA-B11}$VLP$^{h+RNA}$, PVY (PDB ID: 6HXX), $^{PVY}$VLP$^{h+RNA}$ (PDB ID: 8OPC). **e** Structural superposition on all Cα atoms of CP units from RNA-free helical filaments $^{PVA-B11}$VLP$^{h8.5}$ and $^{PVY}$VLP$^{h7.5}$ (PDB ID: 8OPB). In (**c–e**), dashed frames highlight notable structural differences between CPs.

significantly lower rise value (4.74 Å) than $^{PVY}$VLP$^{h7.5}$ (5.88 Å)[14] (Fig. 2b, Table 1, Supplementary Table 1). The $^{PVA-B11}$VLP$^{h8.5}$ filaments are thus wider and more compact than $^{PVY}$VLP$^{h7.5}$, resulting in a larger network of interacting CPs in $^{PVA-B11}$VLPs$^{h8.5}$ than in $^{PVY}$VLPs$^{h7.5}$ (Supplementary Fig. 5a). The $^{PVA-B11}$CP$^{h8.5}$ and $^{PVY}$CP$^{h7.5}$ have a similar fold, with their C-IDRs unstructured due to the absence of RNA[14] (Fig. 2c, e). However, a slight difference in the conformation of the sN-IDR kink (Fig. 2e) likely enables the differences in the architectures of $^{PVA}$VLP$^{h8.5}$ and $^{PVY}$VLP$^{h7.5}$ (Fig. 2b)[14].

As the yield of $^{PVA-B11}$VLPs was relatively low, we tested the production of VLPs by a CP from the PVA isolate Datura, which differs from $^{PVA-B11}$CP in two residues in the uN-IDR (Supplementary Fig. 5b). The use of $^{PVA-Datura}$CP indeed significantly increased the production of VLPs. Since $^{PVA-Datura}$VLPs were structurally identical to $^{PVA-B11}$VLPs (Supplementary Fig. 5c, d, Table 1, Supplementary Table 1), $^{PVA-Datura}$CP was used for further experiments.

$^{PVA}$CP thus forms $^{PVA}$VLPs with a lower degree of structural polymorphism than $^{PVY}$VLPs[14]. While $^{PVA}$VLP$^{h+RNA}$s have a virion-like

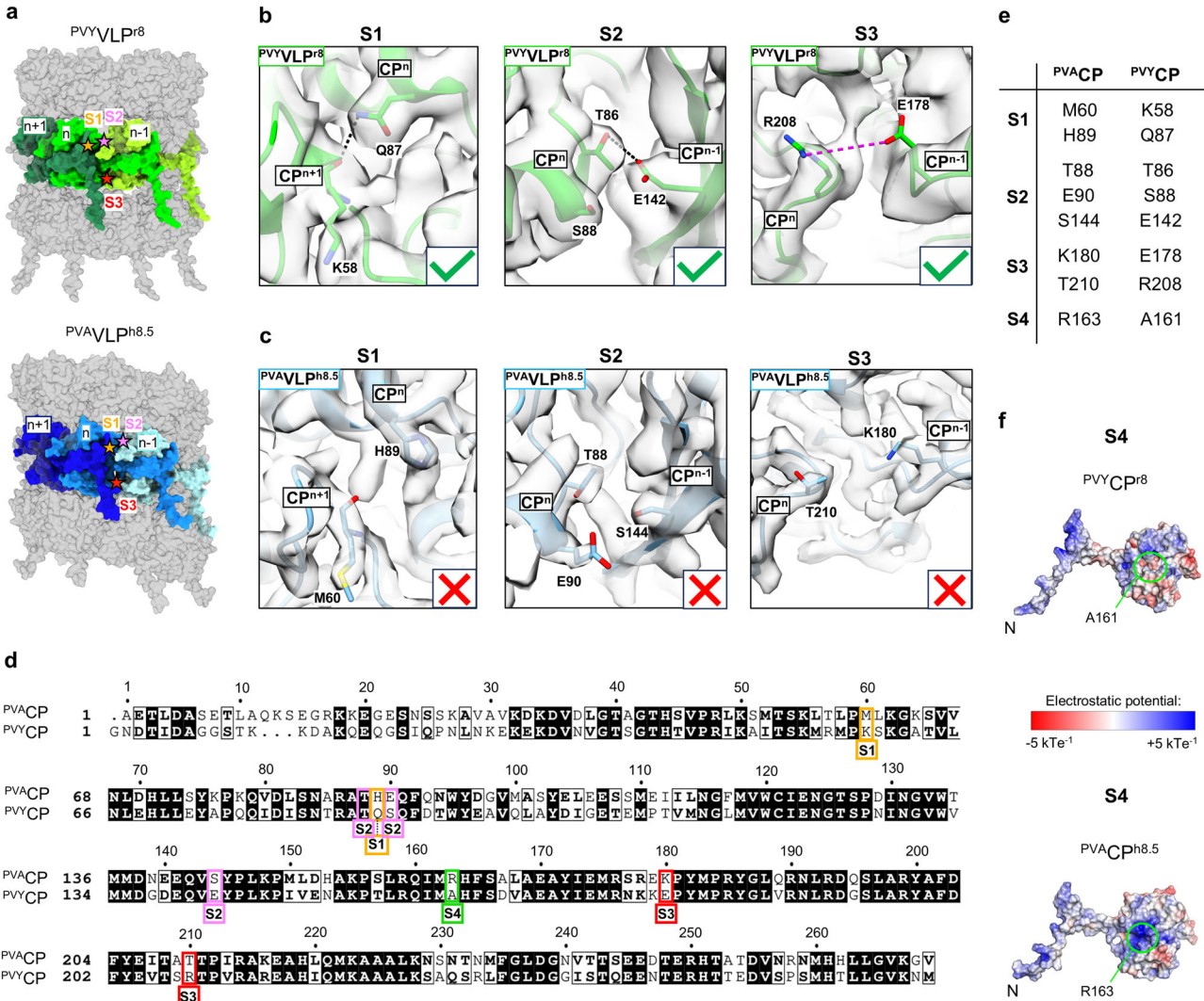

**Fig. 3 | Identification of sites S1-S4 in potyviral VLPs. a** Locations of sites S1-S3 (colored stars) at the CP$^{n+1}$/CP$^n$/CP$^{n-1}$ interfaces in structures of $^{PVY}$VLP$^{r8}$ (PDB ID: 8OPA) and $^{PVA-Datura}$VLP$^{h8.5}$, displayed as surfaces. (CP$^{n+1}$, CP$^n$ and CP$^{n-1}$ are colored in shades of green or blue). **b, c** Close-up views of S1-S3 in the atomic models (ribbon) of $^{PVY}$VLP$^{r8}$ (PDB ID: 8OPA, EMD-17046, in (**b**) and $^{PVA-Datura}$VLP$^{h8.5}$ (**c**), in their corresponding cryo-EM density maps. The amino acid residues of interest are shown in sticks. Hydrogen bonds and salt bridges are presented as dotted lines in black and magenta, respectively. Green tick symbol in (**b**) marks existing interaction; red cross sign in (**c**) marks the absence of interactions. **d** Amino acid sequence alignment of $^{PVA-Datura}$CP (GenBank accession number: Y11426) and $^{PVY}$CP (GenBank accession number: KM396648). Conserved residues: black shade; similar residues: bold text. The numbering of residues is based on the $^{PVA}$CP. Residues in S1-S4 sites are highlighted with colored rectangles. The alignment was created using the MUSCLE algorithm[66]. **e** List of residues building sites S1-S4 in RNA-free $^{PVA-Datura}$CP VLPs and $^{PVY}$VLPs. **f** Location of a site S4 facing the lumen of $^{PVY}$CP$^{r8}$ (PDB ID: 8OPA) and $^{PVA-Datura}$CP$^{h8.5}$, electrostatic surface potential is shown (red: negative charge, blue: positive charge).

architecture[11–14,20], like $^{PVY}$VLP$^{h+RNA}$s[14], the RNA-free helical $^{PVA}$VLPs have distinct helical parameters, with a higher number of CP units per turn and are more compact than those formed by $^{PVY}$CP [14]. Importantly, $^{PVA}$CP does not form RNA-free stacked-ring filaments.

### Identification of candidate CP residues that affect RNA-free VLP architecture

We then searched for potential structural determinants that could explain the differences in VLP architecture of the two potyviral species, in particular the lack of stacked-ring $^{PVA}$VLPs. We analyzed the CP-CP interactions in $^{PVY}$VLP$^{r8}$ (PDB ID: 8OPA) and $^{PVY}$VLP$^{h7.5}$ (PDB ID: 8OPB)[14], focusing on the interactions that are absent in $^{PVA}$VLP$^{h8.5}$ due to the amino acid differences between $^{PVA}$CP and $^{PVY}$CP. Thus, we identified the interaction sites S1, S2, and S3 at the interfaces of the neighboring subunits CP$^{n+1}$, CP$^n$ and CP$^{n-1}$ in RNA-free $^{PVY}$VLPs, formed by the non-conserved residues (Fig. 3a).

In S1 in $^{PVY}$VLP$^{r8}$ (Fig. 3b) (and $^{PVY}$VLP$^{h7.5}$, Supplementary Fig. 6) this includes the side chain of Gln87 (CP$^n$) and the main chain carbonyl of Lys58 (CP$^{n+1}$). The corresponding residues in $^{PVA}$CP are His89 (CP$^n$) and Met60 (CP$^{n+1}$), which do not interact in $^{PVA}$VLP$^{h8.5}$ (Fig. 3c–e). S2 in $^{PVY}$VLP$^{r8}$ (Fig. 3b) (and $^{PVY}$VLP$^{h7.5}$, Supplementary Fig. 6) comprises the side chains of Thr86 ($^{PVY}$CP$^n$) and Glu142 ($^{PVY}$CP$^{n-1}$). In $^{PVA}$VLP$^{h8.5}$, the corresponding residues Thr88 ($^{PVA}$CP$^n$) and Ser144 ($^{PVA}$CP$^{n-1}$) do not interact (Fig. 3c–e). S3 in $^{PVY}$VLP$^{r8}$ (Fig. 3b) (but not in $^{PVY}$VLP$^{h7.5}$, Supplementary Fig. 6) comprises the side chains of Arg208 (CP$^n$) and Glu178 (CP$^{n-1}$). The corresponding residues in $^{PVA}$CP are Thr210 (CP$^n$) and Lys180 (CP$^{n-1}$), which do not interact in $^{PVA}$VLP$^{h8.5}$ (Fig. 3c–e).

To test whether the replacement of $^{PVA}$CP residues forming these sites would trigger the formation of stacked-ring filament, we prepared $^{PVA}$CP$^{H89Q}$ for S1. For S2, we first aimed to prepare $^{PVA}$CP$^{S144E}$. To avoid possible electrostatic repulsion between the side chains of Glu90 ($^{PVA}$CP$^n$) and the new Glu in $^{PVA}$CP$^{n-1}$ (Fig. 3c), we also mutated Glu90 to Ser (Ser88 in

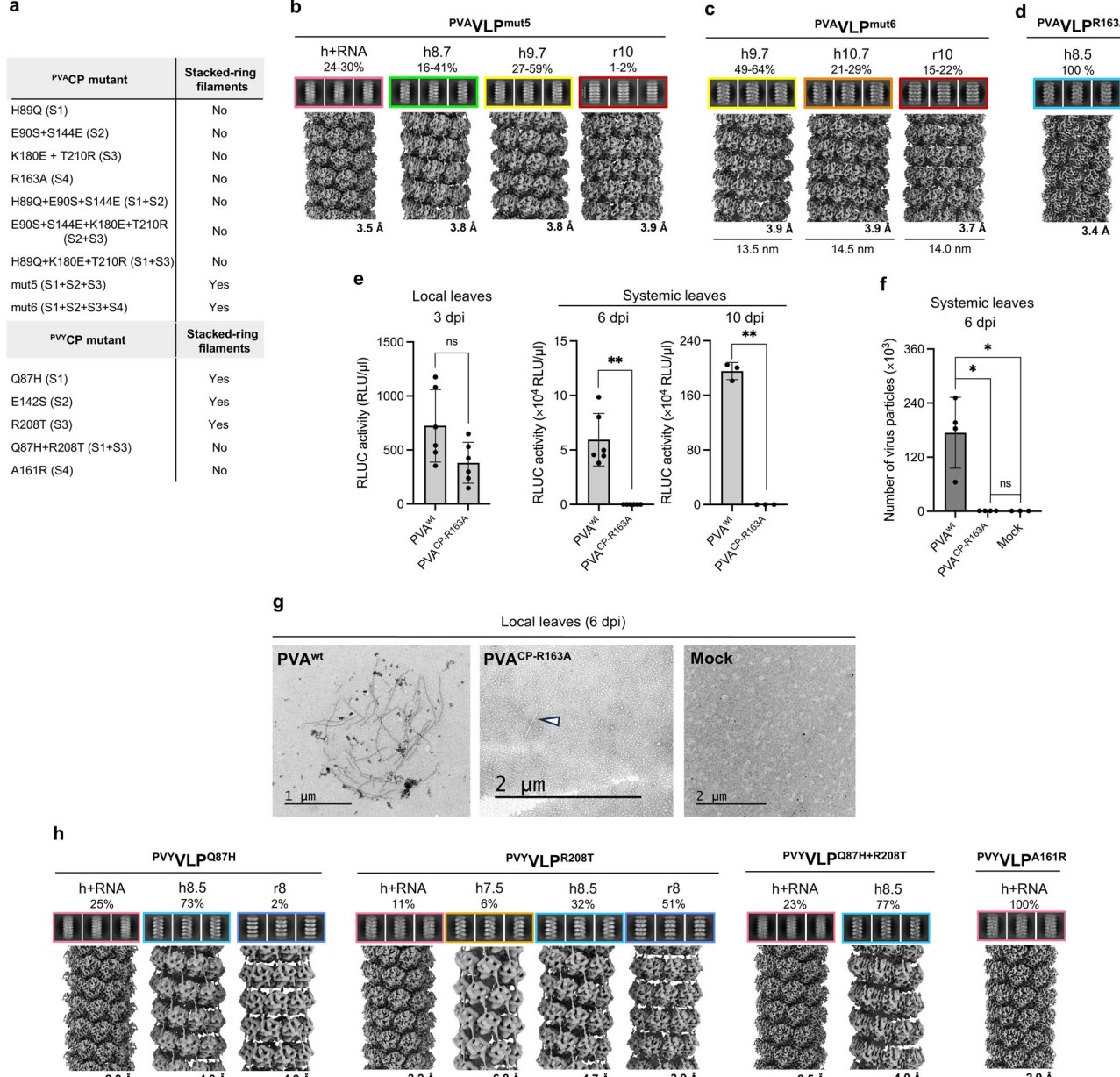

**Fig. 4 | Transformation of VLP architecture and structural polymorphism modulated by mutations at sites S1–S4 in PVACP and PVYCP and importance of S4 site for RNA encapsidation in vitro and in planta. a** List of PVACP and PVYCP mutants and their ability to form stacked-ring VLPs. **b–d** Cryo-EM analysis of PVAVLPmut5 (**b**) PVAVLPmut6 (**c**), and PVAVLPR163A (**d**): 2D class averages (top) with their cryo-EM density maps (bottom). **e** PVAwt and PVACP-R163A gene expression measured as *Renilla* luciferase (RLUC) activity (RLU: relative light unit) in infected *Nicotiana benthamiana* local leaves at 3 dpi (left) and systemic leaves at 6 and 10 dpi (right); *n* = 6 plants (3 and 6 dpi) and *n* = 3 plants (10 dpi samples). **f** PVAwt and PVACP-R163A gene virus particle accumulation in systemically infected *N. benthamiana* leaves at 6 dpi determined by immuno-capture RT-qPCR; *n* = 4 plants (PVAwt and PVACP-R163A) and *n* = 3 plants

(mock). In **e** and **f** error bars (in black) represent standard deviation. Student's t-test or Mann-Whitney U-test were used to calculate statistical significance (ns – *p* > 0.5, * - *p* < 0.05, ** - *p* < 0.01; N = 3–6). Raw data for panels e and f are provided in the Supplementary Data 1 file. **g** Negative stain TEM of PVAwt, PVACP-R163A, and mock samples from the locally infected *N. benthamiana* leaves at 6 dpi. **h** 2D class averages of mutant PVYVLPs (above) with their cryo-EM density maps (bellow). In (**b**-**d** and **h**) overall resolutions of respective cryo-EM maps are marked below in Å. Architecturally distinct filaments are highlighted in pink (h+RNA), green (h8.7), yellow (h9.7), orange (h10.7), red (r10), and light blue (h8.5). The % above 2D classes mark the proportion of VLPs of a certain architecture among all VLPs produced by a specific CP.

PVYCPn), to form PVACPE90S+S144E. To restore the salt bridge at S3, we constructed PVACPK180E+T210R.

The mutants covering a single site, i.e., PVACPH89Q (S1), PVACPE90S+S144E (S2) or PVACPK180E+T210R (S3), did not form stacked-ring filaments nor did their pairwise combinations PVACPH89Q+E90S+S144E (S1 + S2), PVACPE90S+S144E +K180E+T210R (S2 + S3) or PVACPH89Q+K180E+T210R (S1 + S3) (Fig. 4a, Supplementary Fig. 7a–f). However, simultaneous mutations at all three sites, S1 + S2 + S3, i.e. PVACPH89Q+E90S+S144E+K180E+T210R (PVACPmut5), led to the

formation of stacked-ring filaments (PVAVLPmut5:r), in up to 2 % of all filaments formed (Fig. 4a, b, Supplementary Figs. 7g, 8a, Table 1). Interestingly, in contrast to the octameric rings in PVYVLPr8, the rings formed by PVAVLPmut5 were decameric (PVAVLPmut5:r10) (Fig. 4b, Table 1, Supplementary Table 1). PVACPmut5 also formed the virion-like PVAVLPmut5:h+RNAs and two types of RNA-free left-handed helical assemblies, with 8.7 CPs/helical turn (PVAVLPmut5:h8.7) or 9.7 CPs/helical turn (PVAVLPmut5:h9.7). The latter has a

more compact structure due to a lower helical rise compared to those with 8.5/8.7 CPs/helical turn (Fig. 4b, Table 1, Supplementary Table 1).

As $^{PVY}$CP preferentially forms stacked-ring filaments[14], we then searched for other non-conserved residues whose replacement in $^{PVA}$CP with the corresponding $^{PVY}$CP residues would increase the proportion of the stacked-ring filaments. We found that the filament lumen of $^{PVA}$VLP$^{h8.5}$ has a significantly more positive electrostatic surface potential than $^{PVY}$VLP$^{r8}$ (and $^{PVY}$VLP$^{h7.5}$) mainly due to Arg163 in $^{PVA}$CP (Fig. 3f, Supplementary Fig. 6). In $^{PVY}$CP this Arg is replaced with Ala161 (Fig. 3d–f). Arg163 in $^{PVA}$CP forms a hydrogen bond with the carbonyl oxygen of Ala85 of the same CP in $^{PVA}$VLP$^{h8.5}$ (Supplementary Fig. 8b), and such interaction is not possible in $^{PVY}$VLP$^{r8}$/$^{PVY}$VLP$^{h7.5}$ due to Ala161. This interaction in $^{PVA}$VLP$^{h8.5}$ might limit the flexibility of the sN-IDR and consequently affect VLP architecture and structural polymorphism. We labeled this site as S4 and mutated Arg163 to Ala in $^{PVA}$CP$^{mut5}$ to form $^{PVA}$CP$^{H89Q+E90S+S144E+K180E+T210R+R163A}$ ($^{PVA}$CP$^{mut6}$).

Replacement of Arg163 with Ala at S4 indeed increased the formation of stacked-ring filaments by $^{PVA}$CP$^{mut6}$ to 15–22%, forming decameric rings ($^{PVA}$VLP$^{mut6:r10}$) (Fig. 4a, c, Supplementary Fig. 7h, 8a, c), similar to $^{PVA}$VLP$^{mut5:r10}$ (Fig. 4b, Table 1, Supplementary Table 1). $^{PVA}$CP$^{mut6}$ also formed two wide types of RNA-free filaments with left-handed helical symmetry, with 9.7 CPs/turn ($^{PVA}$VLP$^{mut6:h9.7}$) or 10.7 CPs/turn ($^{PVA}$VLP$^{mut6:h10.7}$) (Fig. 4c, Table 1, Supplementary Table 1).

$^{PVA}$CP$^{mut6}$ completely lost the ability to form VLP$^{h+RNA}$ filaments, which we attribute to the Arg163Ala mutation, as Arg163 is involved in the binding of RNA in PVA and $^{PVA}$VLP$^{h+RNA}$ (Fig. 1f, Supplementary Fig. 8d). This was confirmed by the single mutant $^{PVA}$CP$^{R163A}$, which produced only helical RNA-free filaments (Fig. 4d, Supplementary Fig. 8a, Table 1).

In all three types of VLPs formed by $^{PVA}$CP$^{mut6}$ (Supplementary Fig. 9a), the mutated residues facilitated the formation of novel CP-CP contacts (Supplementary Fig. 9b), corresponding to those found in S1–S3 sites in $^{PVY}$VLP$^{r8}$ (Fig. 3b) and $^{PVY}$VLP$^{h7.5}$ (Supplementary Fig. 6). In addition, we found a salt bridge between the new Arg210 in $^{PVA}$CP$^{mut6}$ (CP$^{n}$) and Asp152 in CP$^{n-10}$ in $^{PVA}$VLP$^{mut6:h9.7}$ (CP$^{n-11}$ in $^{PVA}$VLP$^{mut6:h10.7}$; CP$^{m-1}$ in $^{PVA}$VLP$^{mut6:r10}$) (Supplementary Fig. 9b).

Although $^{PVA}$CP$^{mut6}$ forms three architecturally distinct types of VLPs, their CP-CP interaction networks are very similar (Supplementary Fig. 9c), as are the structures of their $^{PVA}$CP$^{mut6}$ units (Supplementary Fig. 9d). The loop Ser127-Gly132 in all VLPs formed by $^{PVA}$CP$^{mut6}$ expectedly adopts the conformation found in RNA-free VLPs (Fig. 2c). Subtle structural differences can be found in their N-IDR kink regions (His44-Gly63), which likely enable the structural polymorphism of RNA-free $^{PVA}$VLP$^{mut6}$s. Surprisingly, we observed that the angle of their sN-IDR kink adopts the conformation found in viruses and VLP$^{h+RNA}$ and not the one typical for RNA-free filaments (Supplementary Fig. 9d).

To summarize, the non-conserved residues, mainly from the globular CP core region, determine the architecture of RNA-free VLPs and the degree of their structural polymorphism. In $^{PVY}$CP, interactions between residues at sites S1–S4 support the formation of RNA-free stacked-ring filaments, which cannot be formed by $^{PVA}$CP due to a different amino acid sequence. The non-conserved residue Arg163 at S4 in $^{PVA}$CP not only inhibits the formation of stacked rings but is also essential for the formation of $^{PVA}$VLP$^{h+RNA}$. Interestingly, the $^{PVA}$CP$^{mut5}$ and $^{PVA}$CP$^{mut6}$ constructs, which do not occur in nature, assemble into the RNA-free filaments with a higher number of CP units per turn and a more compact architecture than those formed by wild-type $^{PVA}$CP.

### Arg163 in $^{PVA}$CP is essential for successful PVA infection

We next assessed the impact of the $^{PVA}$CP$^{R163A}$ mutation on the PVA infection *in planta* and introduced the change in the PVA infectious cDNA. Three days post-infection (3 dpi), virus-derived reporter expression in a local PVA$^{CP-R163A}$ infection was similar to the wild-type control infection suggesting the mutation did not interfere with viral protein production (Fig. 4e). In contrast, no PVA$^{CP-R163A}$-derived reporter activity was detected in the systemic leaves at 6 and 10 dpi and particle quantity was at background level at 6 dpi (Fig. 4f). No full-length virus particles were observed in the PVA$^{CP-R163A}$-infected local leaf samples (Fig. 4g). The rare particles observed were probably RNA-free VLPs, as shown for $^{PVA}$VLP$^{R163A}$ (Fig. 4d).

Thus, the PVA$^{CP-R163}$ mutation did not interfere with viral gene expression in the local agroinfiltrated leaves but disabled both virion particle production and systemic spread of the virus in *N. benthamiana*.

### The replacement of S1-S4 residues in $^{PVY}$CP with those of $^{PVA}$CP confirms their structural role

To further prove the role of S1-S4 residues, we replaced them in $^{PVY}$CP with those from $^{PVA}$CP. Cryo-EM analysis of the purified VLPs formed by $^{PVY}$CP$^{Q87H}$ (S1) showed a significant drop in stacked-ring filament production, as they represented only 2 % of all VLPs (Fig. 4h, Supplementary Fig. 10a, Table 1, Supplementary Table 1) compared to 65 % formed by wild-type $^{PVY}$CP[14]. $^{PVY}$CP$^{Q87H}$ still formed the virion-like VLP$^{h+RNA}$s, as well as the RNA-free helical filaments (Fig. 4h). However, the number of CP units per helical turn of RNA-free helical filaments increased from 7.5 formed by the wild-type $^{PVY}$CP to 8.5 in $^{PVY}$VLP$^{Q87H:h8.5}$ (Table 1, Supplementary Table 1).

Filament formation by $^{PVY}$CP$^{E142S}$ (S2) was poor, suggesting that the interaction of Thr86 (CP$^{n}$) with Glu142 (CP$^{n-1}$) is important for the stability of $^{PVY}$VLPs. Although we still observed the stacked-ring filaments (Supplementary Fig. 10b), the instability of the VLPs prevented detailed cryo-EM analysis.

The substitution in S3 ($^{PVY}$CP$^{R208T}$) resulted in only a slight reduction in the population of the stacked-ring filaments, to about 50 % of all filaments. $^{PVY}$CP$^{R208T}$ formed two types of RNA-free helical filaments. A small portion retained parameters close to those of wild-type $^{PVY}$CP ($^{PVY}$VLP$^{h7.5}$), while the larger population had 8.5 CPs/turn, with the helical rise value between $^{PVY}$VLP$^{h7.5}$ and $^{PVA}$VLP$^{h8.5}$. $^{PVY}$CP$^{R208T}$ also formed VLP$^{h+RNA}$s (Fig. 4h, Supplementary Fig. 10a, Table 1, Supplementary Table 1).

Finally, the double mutant $^{PVY}$CP$^{Q87H+T208R}$ (S1 + S3) lost the ability to form stacked-ring filaments (Fig. 4h, Supplementary Fig. 10a). It formed virion-like VLP$^{h+RNA}$s and RNA-free helical filaments of 8.5 CPs/turn (Fig. 4h, Table 1, Supplementary Table 1).

The mutation in S4, $^{PVY}$CP$^{A161R}$, prevented the formation of stacked-ring filaments and led exclusively to the formation of virion-like particles (Fig. 4h, Supplementary Fig. 10a, Table 1, Supplementary Table 1). This emphasizes the importance of S4 for the formation of VLP$^{h+RNA}$ and stacked-ring filaments.

These experiments confirm the importance of residues at sites S1-S4 for the architecture, structural polymorphism and stability of VLPs. The contribution of individual residues depends on the amino acid context of the specific CP. Again, the unnatural mutants of $^{PVY}$CP tend to form RNA-free VLPs with a higher number of CP units per helical turn or ring than wild-type $^{PVY}$CP (Supplementary Table 1).

### Architecture of VLPs and degree of their structural polymorphism are species-specific with similarities within the potyvirus clades

Next, we expanded the pool of potyviral CPs. Based on the phylogenetic tree of potyviruses (https://ictv.global/system/files/inline-images/OPSR.Pot_.Fig3_.v15.png)[26], we selected pepper mottle virus (PepMoV; *Potyvirus capsimaculae*) from the PVY clade, tobacco etch mosaic virus (TEV; *Potyvirus nicotianainsculpentis*) as a clade mate of PVA, and Johnsongrass mosaic virus (JGMV; *Potyvirus halapensis*) as a member of a distant clade. Similar clade distribution of potyviruses was obtained based on the CP sequences (Fig. 5a). We found limited variability of residues at sites S1-S4 between potyvirus species (Fig. 5b, c, Supplementary Fig. 11). This variability is even lower between members of the same clade, especially in the PVY clade, and converges to high conservation in strains or isolates of the same species (Supplementary Fig. 12).

While 150 nm to 3000 nm long filaments were detected in the bacterial lysate when $^{TEV}$CP was expressed, they seemed to break down into shorter filaments (20–200 nm) during purification and possibly even denatured (Supplementary Fig. 13a-e). Therefore, cryo-EM analysis of the $^{TEV}$VLPs

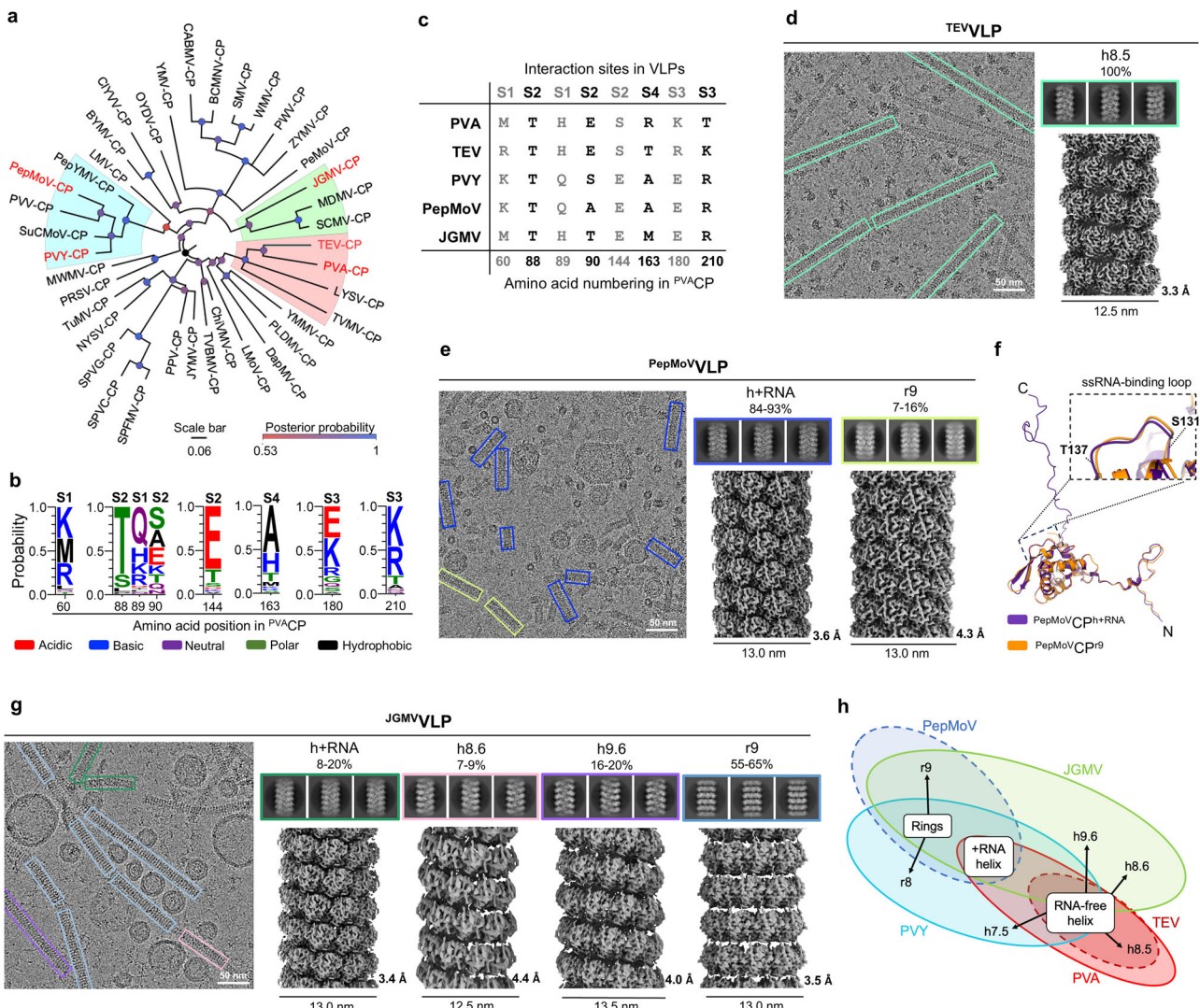

**Fig. 5 | Structural diversity of VLPs across the *Potyvirus* genus. a** Phylogenetic tree showing the relationships among 39 selected potyviral CP sequences (GenBank accession numbers are in the Supplementary Data 1 file) generated by PhyML maximum-likelihood analysis[68]. PVA-B11CP, PVYCP, TEVCP, PepMoVCP, and JGMVCP are highlighted in red. PVA/TEV clade is shaded in red, PVY/PepMoV in blue, and the clade including JGMV in green. Scale bar represents the number of nucleotide changes per site along the branch. **b** Sequence logos (created with WebLogo3[69]) depicting probabilities of amino acid residue type at the sites S1-S4, based on the alignment of potyviral CP sequences used in (**a**) Residue height represents residue probability. **c** Amino acid alignment of S1-S4 residues from PVACP, PVYCP, TEVCP, PepMoVCP, and JGMVCP. **d** Cryo-EM analysis of TEVVLP: cryo-EM micrograph (left), 2D class averages (top right), and cryo-EM density map (bottom right). VLPh8.5 filaments are highlighted in green. **e** Cryo-EM analysis of PepMoVVLPs: cryo-EM micrograph (left), 2D class averages representing the VLPh+RNA and VLPr9 (top right) with cryo-EM maps (bottom right). Architecturally distinct filaments are high-lighted in dark purple (VLPh+RNA) and yellow (VLPr9). **f** Structural superposition on all Cα atoms of PepMoVCPh+RNA and PepMoVCPr9. **g** Cryo-EM analysis of JGMVVLPs: cryo-EM micrograph (left), 2D class averages representing the VLPh+RNA, VLPh8.6, VLPh9.6, and VLPr9 (top right) with their final cryo-EM maps (bottom right). Architecturally distinct filaments are highlighted in green (VLPh+RNA), pink (VLPh8.6), purple (VLPh9.6), and blue (VLPr9). In (**d, e, g**) Diameters of filaments are shown below the cryo-EM density maps in nm and the overall resolution in Å. The % above 2D classes mark the proportion of VLPs of a certain architecture among all VLPs produced by a specific CP. **h** Graphical summary of the VLP architectures formed by CPs from PVA, PVY, TEV, PepMoV and JGMV.

was performed on the *E. coli* lysate (Fig. 5d, Supplementary Fig. 13e). Interestingly, only monomorphic filaments with an RNA-free left-handed helical symmetry architecture with 8.5 TEVCPs/turn could be observed (TEVVLPh8.5), with helical parameters similar to those of the clade relative PVAVLPh8.5 (Fig. 5d, Table 1, Supplementary Table 1). The fold of TEVCPh8.5 is similar to that of PVACPh8.5 and even PVYCPh7.5, with slight variations in the N-IDR at residues Thr38-Gln52 (TEVCP numbering) (Supplementary Fig. 14a). The C-terminal α8-helix in TEVCPh8.5, facing the filament lumen, is approximately one turn longer than the α8-helix in PVACPh8.5 or PVYCPh7.5 and continues into a loop (residues Asn224 to Asp233), which folds back onto helices α6 and α8 (Supplementary Fig. 14a) and interacts with them. The tip of this C-terminal loop in TEVCP points towards the canonical RNA-

binding site, which may prevent the RNA binding (Supplementary Fig. 14b). This configuration of TEVCP appears to be energetically more favorable to the one that would bind RNA, as no TEVVLPh+RNAs were observed. The S4 residue in TEVCP, Thr158 (Fig. 5c), could additionally weaken the affinity for RNA.

Most of the filaments formed by PepMoVCP (84–93%, Table 1) exhibited a conserved virion-like architecture (PepMoVVLPh+RNA) (Fig. 5e, Table 1, Supplementary Table 1), and the fold of PepMoVCPh+RNA is similar to that of PVYCPh+RNA (Supplementary Fig. 14c). The purified filaments formed by PepMoVCP were relatively short (Supplementary Fig. 13a, b)[14]. The rest of the filaments (7–16 %, Table 1, Supplementary Table 1) were RNA-free with the stacked-ring architecture. However, in contrast to the octameric rings

formed by its clade mate [PVY]CP, the rings in the [PepMoV]CP-derived filaments are nonameric ([PepMoV]VLP[r9]) (Supplementary Fig. 14d). Moreover, the stacking between the rings in the filament [PepMoV]VLP[r9] is significantly tighter (helical rise of 33.50 Å) compared to [PVY]VLP[r8] (helical rise of 43.42 Å)[12,14] (Supplementary Fig. 14d, Table 1). No [PepMoV]VLP[h] were observed.

In [PepMoV]CP[r9] the conformations of the sN-IDR and that of the canonical RNA-binding site are more similar to those in [PepMoV]CP[h+RNA] than to the CP fold in RNA-free filaments (Fig. 5f, Supplementary Fig. 14e). This could explain the denser packing of CP units along the long axis in [PepMoV]VLP[r9] compared to [PVY]VLP[r8] (Supplementary Fig. 14d)[14].

The length of the purified [JGMV]VLPs varied between 100 to 2000 nm (Supplementary Fig. 13a, b). Their cryo-EM analysis revealed four architecturally distinct filament types with unique structural parameters (Fig. 5g, Table 1, Supplementary Table 1). Three of them exhibit left-handed helical symmetry, the virion-like form [JGMV]VLP[h+RNA], and two RNA-free forms with 8.6 or 9.6 CP units per helical turn ([JGMV]VLP[h8.6] and [JGMV]VLP[h9.6]). The fourth and most abundant filament type has a stacked-nonameric-ring architecture ([JGMV]VLP[r9]) (Fig. 5g, Supplementary Fig. 14d–g, Table 1, Supplementary Table 1).

The helical parameters of RNA-free [JGMV]VLP[h8.6] are similar to those of [PVA]VLP[h8.5] and [TEV]VLP[h8.5] (Table 1). The overall structure of [JGMV]CP[h8.6] resembles CP[h]s from other potyviruses, except for a notable divergence in the angle of the of the sN-IDR kink, residues Phe79-Lys90 (Supplementary Fig. 14g). [JGMV]VLP[h9.6] represents another architectural type of RNA-free helical filaments among wild-type potyviral VLP[h] with 9.6 [JGMV]CP units per turn (and a helical rise of 4.50 Å), resulting in a larger filament diameter (Fig. 5g, Table 1). Despite significant differences in the structural parameters of [JGMV]VLP[h8.6] and [JGMV]VLP[h9.6], the folds of the subunits, [JGMV]CP[h8.6] and [JGMV]CP[h9.6], are very similar (Supplementary Fig. 14g), with only subtle differences in the fold of their N-IDRs.

[JGMV]VLP[r9] is composed of nonameric stacked rings with the helical rise and twist values similar to those of [PVY]VLP[r8] (Supplementary Fig. 14d, Table 1)[12,14]. The different ring stoichiometries are likely due to the structural plasticity of sN-IDR (Supplementary Fig. 14e).

The residues at sites S1-S3 in [TEV]VLP[h8.5] do not interact (Fig. 5c, Supplementary Fig. 15a), similar to the RNA-free VLPs of the clade-related [PVA]CP and in contrast to the ones formed by more distant [PVY]CP (Figs. 3b, c, 5a–c). In [PepMoV]CP, S1, S3 and S4 residues are conserved compared to [PVY]CP, except for Ala94 of S2 in [PepMoV]CP, which is Ser88 in [PVY]CP (Fig. 5c). Despite the limited resolution of cryo-EM density map in [PepMoV]VLP[r9], the positioning of the residues in the structural model suggests the possibility that the residues at sites S1 and S3 interact similarly to their counterparts in [PVY]VLP[r8] (Fig. 3b, Supplementary Fig. 15b). [JGMV]VLP[r9] shows conserved contacts at sites S2 and S3 compared to [PVY]VLP[r8] (Fig. 3b, Supplementary Fig. 15c). In contrast, the interaction at S1 in [JGMV]VLP[r9] is absent, with the residues composing this site the same as found in [PVA]VLP[h8.5], and the residue at S4 is Met198 (Fig. 5c, Supplementary Fig. 15c).

Thus, although the structures of potyviral VLPs are species-specific, we found certain similarities between members of the same clade (Fig. 5h). Clade mates [PVA]CP and [TEV]CP share the architecture of the RNA-free helical VLP[h8.5] and do not form stacked-ring filaments due to the absence of interactions at S1-S3 sites, and S4 residue restricts the flexibility of sN-IDR. However, Thr158 in S4 of [TEV]CP may be responsible for the absence of [TEV]VLP[h+RNA]s, which are readily formed by [PVA]CP with Arg163 in S4. On the other hand, the conserved residues at the S1 and S3 sites in [PVY]CP and [PepMoV]CP of the PVY clade, support CP-CP interactions and, together with the conserved Ala at S4, enable the formation of stacked-ring VLPs. However, their geometrical parameters and ring stoichiometries are different, likely due to sequence-specific sN-IDR conformations. [JGMV]CP, a member of a separate clade, appears to combine features of the PVY and TEV clades, as it forms stacked-ring filaments supported by interactions at S2 and S3, and slightly wider helical RNA-free filaments, respectively.

## The structural role of the highly conserved triad of charged residues at the CP-CP interface

Previously we have shown that PVY and its VLPs are stabilized by the interactions between the three charged, highly conserved residues (Supplementary Fig. 1b) Arg46 from the sN-IDR (CP[n]), Asp136 and Glu139 from the β-hairpin of the CP[n-10] core[14]. Individual replacement of these residues with Cys or Ala largely favored the formation of VLP[h+RNA]s[14]. In [PVA]CP, the corresponding residues are Arg48, Asp138 and Glu141 (Fig. 3e), and replacement of Asp138 in [PVA]CP with Cys ([PVA]CP[D138C]) also resulted in monomorphic, virion-like filaments, [PVA]VLP[D138C:h+RNA] (Fig. 6a, Table 1, Supplementary Table 1), similar to those formed by [PVY]CP[D136C] [14], and no RNA-free VLPs could be observed.

Thus, in contrast to the species-specific structural features in virions and VLPs, the highly conserved triad of charged residues at the CP-CP interface is crucial for the integrity of virions and even more so for the stability of RNA-free VLPs. The high conservation of this triad in potyviral CPs suggests that its presence is required for the CP to withstand evolutionary pressure. Its crucial role in the formation of RNA-free VLPs suggests that highly ordered RNA-free CP oligomers, with structural parameters similar to those of species-specific VLPs, may act as intermediates in the viral infection cycle.

## Potyviral CP has no ATPase activity when packaged in virions and VLPs or as a monomer

In a previous study, ATPase activity of [PVA]CP produced in *E. coli* was reported, suggesting that ATP binding and hydrolysis are required for the infection steps involving CP[27].

We tested the ATPase activity of different forms of potyviral CP, packaged in virions and VLPs, and as monomeric CP units. Previously, we had shown that the correctly folded monomeric [PVY]CP can be produced as a fusion with the maltose-binding protein (MBP) at its C-terminus ([PVY]CP-MBP)[14]. This prevented the self-assembly of recombinant [PVY]CP into VLPs in bacteria. Subsequent proteolytic removal of MPB from the purified monomeric [PVY]CP-MBPs led to the formation of VLPs in vitro[14]. Here, monomeric [PVA]CP-MBP was produced in a similar manner. Proteolytic cleavage of MBP triggered the formation of monomorphic [PVA]VLP[h8.5]s in vitro (Fig. 6b, Supplementary Fig. 16a, b, Table 1, Supplementary Table 1). Therefore, [PVY]CP-MBP (Supplementary Fig. 16a, c) and [PVA]CP-MBP were used in the ATPase assay.

Purified PVA showed no ATPase activity, but we detected significant ATPase activity in both the [PVA]VLP and [PVY]VLP samples (Fig. 6c, Supplementary Fig. 17). We used an ATPase p97[28] as a positive control and bovine serum albumin (BSA) and the sample buffer with ATP/Mg$^{2+}$ as negative controls. In negative stain TEM micrographs of the VLP samples, we observed vesicle-like objects that were not present in the PVA sample (Fig. 6d, Supplementary Fig. 2b). We then prepared the *E. coli* lysate without the CP overexpression, which was purified using the same protocol as for VLPs. The negative stain TEM imaging of these control samples showed the same type of vesicle-like impurities as in the VLP samples and strong ATPase activity (Fig. 6c, d, Supplementary Fig. 17). This suggests that the ATPase activity may originate from the bacterial background. To test this, we included an additional step in the VLP purification procedure that successfully removed vesicle-like impurities (Fig. 6d) using heparine sepharose chromatography. The purified VLPs (VLP[clean]) had no ATPase activity (Fig. 6c, Supplementary Fig. 17). Purified monomeric forms of [PVA]CP-MBP and [PVY]CP-MBP, which were free of vesicle-like particles from bacteria also showed no ATPase activity (Fig. 6d, Supplementary Fig. 17).

SDS-PAGE analysis of the [PVA]VLP and [PVY]VLP samples revealed a band just below 40 kDa, which was not present in [PVA/PVY]VLP[clean] samples (Supplementary Fig. 17a). N-terminal sequencing and mass spectrometry (MS) of this band in the [PVA]VLP sample (Fig. 6e, Supplementary Table 2) identified OmpF, a protein associated with the outer membrane of *E. coli*[29]. MS analysis also revealed trace amounts of *E. coli* proteins involved in ATP-hydrolysis, such as OppD and OppF of the ABC transporter complexes OppABCDF and MppA-OppBCDF[30], MalK of the ABC transporter

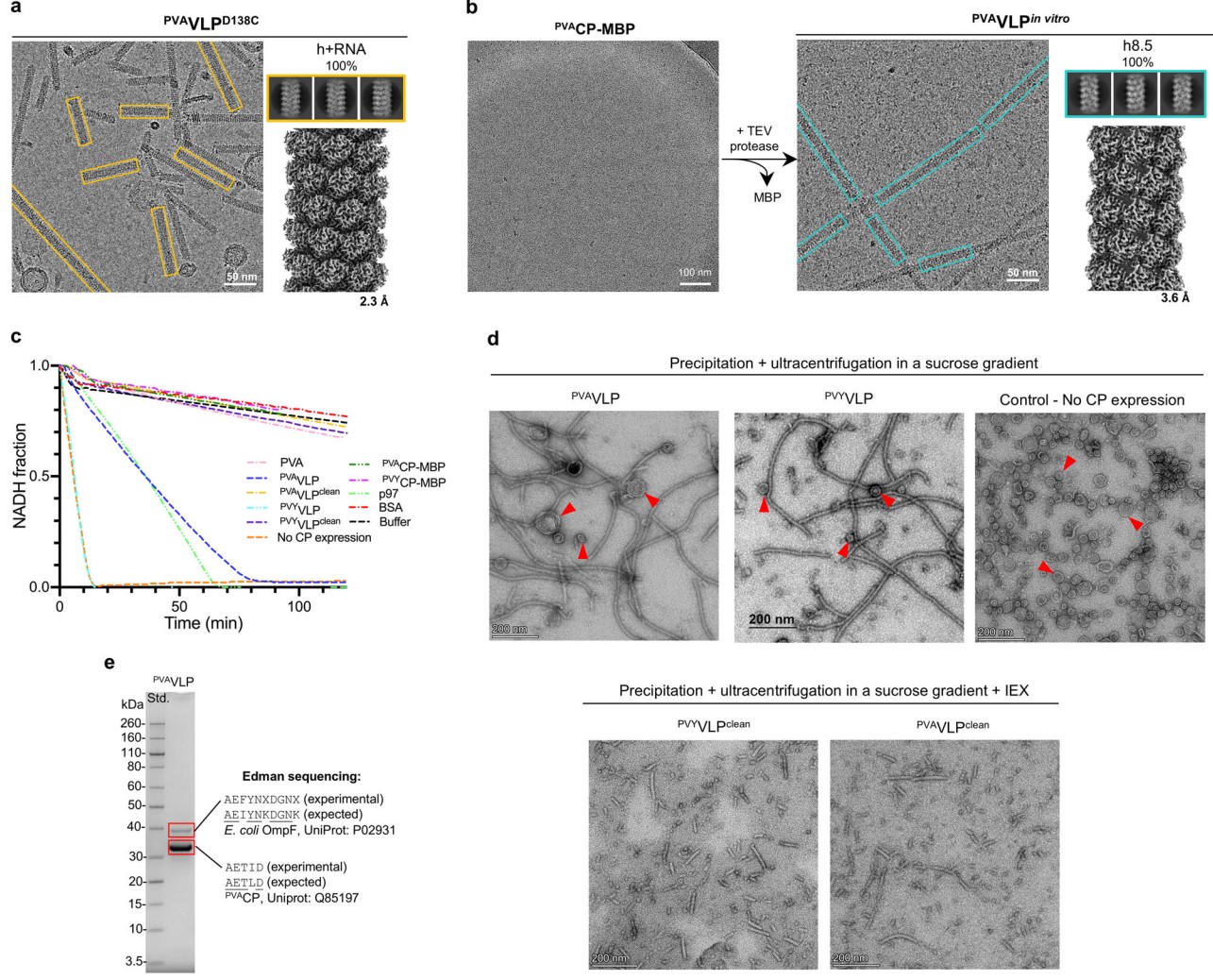

**Fig. 6 | Importance of the highly conserved triangle of the charged residues and the absence of ATPase activity in the CP. a** Cryo-EM micrograph (above left), 2D class averages (top right), and cryo-EM density map (bottom right) of $^{PVA}VLP^{D138C}$. **b** In vitro assembly of $^{PVA}VLPs$ from the monomeric $^{PVA}CP-MBP$ fusion. Left: cryo-EM micrograph of $^{PVA}CP-MBP$. Right: Cryo-EM micrograph (left), 2D class averages (top right), and cryo-EM density map (bottom right) of $VLP^{h8.5}$ formed by $^{PVA-B11}CP$ after the removal of MBP with the TEV protease. In panels (**a** and **b**), the overall resolution is shown in Å. **c** ATPase activity of PVA, different VLPs and CP-MBPs, shown as depletion of NADH in ATP/NADH coupled ATPase assay[52]. Positive control: ATPase p97; negative control: bovine serum albumin (BSA). For clarity, only representative measurements are shown. Each measurement was repeated 3 times (see Supplementary Fig. 17b). Data for curves are provided in the Supplementary Data 1 file. **d** Negative stain TEM analysis of $^{PVA}VLPs$, $^{PVY}VLPs$, and control sample - *E. coli* lysate in the absence of CP expression, purified using the VLP purification protocol (precipitation and ultracentrifugation in a sucrose gradient). $^{PVA}VLP^{clean}$ and $^{PVY}VLP^{clean}$ are VLP samples additionally purified over Heparin Sepharose 6 fast-flow column to remove impurities. Red arrows indicate circular *E. coli* impurities. **e** SDS-PAGE analysis of $^{PVA}VLP$ sample (left) and results of N-terminal sequencing of protein bands (right). Uniprot IDs of identified proteins are added. 'X' marks the amino acid residue whose identity could not be determined.

---

complex MBP–MalFGK2[31,32], and an ATP-dependent recombinase RecA[33] (Supplementary Table 2).

Thus, under conditions used in this study, potyviral CP has no ATPase activity when packaged in virions and VLPs or as a monomer.

## Discussion

Potyviruses are the largest group of known plant viruses and represent a major economic burden worldwide[1,2,21,34,35]. Their dynamic evolution and complex biology are influenced by agriculture, trade and exposure to new hosts and aphid vectors[2,21,36]. Their virions must be sturdy enough to protect the RNA genome *ex-planta* and during the transfer via their biological vectors to healthy host plants. They must also be labile enough to initiate infection as they enter new cells. Identifying the factors that drive particle assembly, disassembly and stability is central to understanding virus biology, evolution and control. Our results suggest that while potyviruses share the conserved quaternary structure of the virion, the amino acids forming the CP-CP and CP-RNA interactions leading to this structure are species-

specific. This is also reflected in the subtle structural differences in the CP units that form the capsids between different potyvirus species and virion-like $VLP^{h+RNA}s$. These include the species-specific residues in both extended regions of the CP, the sN-IDR, involved in CP-CP interactions, and two sites in the C-IDR that contribute to CP-RNA interactions (Fig. 1c–f, Supplementary Figs. 3, 11).

Second, the species-specific structural features are even more pronounced in the RNA-free VLPs produced by potyviral CPs. In this case, the species-specific residues from the globular CP core region together with the sN-IDR, located in sites S1-S4, are responsible for the species-specific architectures of the RNA-free VLPs and the unique structural features in their CP units, as well as for the degree of their structural polymorphism (Figs. 2–5, Supplementary Figs. 9, 14, 15). The chemical nature of the CP core residues forming sites S1-S4 appears important, as mutations at these positions significantly alter the architectural parameters of RNA-free VLPs compared to those produced by wild-type CPs (Figs. 2–5, Table 1). This suggests that the $^{PVA}CP$ and $^{PVY}CP$ mutants used in this study (Fig. 4a),

which are not present in nature, may lead to aberrant virion assembly. Consistent with the amino acid restrictions at S1-S4 sites, we observed structural similarities in the RNA-free architectures between members of the same clade, although the CPs from their VLPs still exhibit species-specific structural features (Fig. 5, Supplementary Fig. 14, Table 1).

Third, based on the structural and mutational analysis of [PVA]VLPs, we found that Arg163 in [PVA]CP, which in the canonical RNA binding site (site S4, Fig. 1f, 3d, Supplementary Fig. 8d), strongly affects the RNA-binding affinity of [PVA]CP (Fig. 4d). The introduction of the mutation from Arg163 to Ala in the *CP* gene of PVA, PVA[CP-R163], *in planta* showed that although the mutation did not affect viral gene expression in the local agroinfiltrated leaves, it prevented both particle production and systemic spread of the virus in *N. benthamiana* (Fig. 4e–g). This significant biological implication emphasizes the role and importance of the Arg163 residue for the assembly of stable and functional PVA particles, the severe consequences of mutations at this site, and the high validity of virion-like VLP[h+RNA]s as structural models for potyviruses.

Fourth, although sN-IDR exhibits high structural plasticity, we observed that its kink angle can only assume a few discrete values (Fig. 2c, Supplementary Fig. 14e, g). For example, in PVA and PVY and their respective VLP[h+RNA]s, the sN-IDR kink angle is around 50° ('closed') and it increases to around in 70–80° ('open') in RNA-free VLPs (Supplementary Fig. 18a). Interestingly, the highly conserved triad of charged residues appears to hold the sN-IDR kink in place (Supplementary Fig. 1b, 18b). In PVA (and [PVA]VLP[h+RNA]) Arg48 from the sN-IDR in the [PVA]CP[n] subunit is stabilized by the conserved residues Asp138 and Glu141 of the β-hairpin in the core of the adjacent CP[n-10] unit (Supplementary Fig. 18b). In addition, the sN-IDR kink of [PVA]CP[n] at residues Leu56-Val66, where it makes a sharp turn, is bound to the core of [PVA]CP[n-1], where the start of the helix α2 (residues His89-Tyr103) is stabilized by interactions with the residues of the β-hairpin of [PVA]CP[n-2] (Supplementary Fig. 18a). In RNA-free [PVA]VLP[h8.5], the sN-IDR kink is stabilized by similar interactions, despite the larger angle of the sN-IDR kink (Supplementary Fig. 18a). A similar observation can be made for PVY and its VLPs (Supplementary Fig. 18a, b), as well as for other VLPs studied here. Interestingly, however, in case of [PepMoV]VLPs, the sN-IDR angle remains the same in the presence and absence of RNA (Fig. 5f), while in [JGMV]VLPs, the sN-IDR kink adopts two distinct angles in RNA-free VLPs (Supplementary Fig. 14e, g). As we have shown previously for [PVY]CP[14] and here for [PVA]CP (Fig. 6a) by mutating one of the highly conserved residues of this charged triad, we completely prevent the formation of RNA-free VLPs, while strong CP-RNA interactions rescue the formation of VLP[h+RNA]s. This suggests that only those potyviral CPs survive evolutionary selection in which this charged Arg/Asp/Glu triad is conserved, as it is required both for the structural support of the capsid with bound RNA and for RNA-free structurally ordered assemblies of CP with similar structural parameters as in RNA-free VLPs.

Previously we proposed that in the late stage of potyvirus infection, when the cell is saturated with CP, CP molecules or RNA-free oligomeric CP intermediates bind to RNA to release the ribosomes from it, and the free viral RNA can be encapsidated into virions[37]. We now show that in RNA-free CP-assemblies (VLPs or smaller ordered oligomers with similar structural parameters), species-specific CP-CP interactions play a central role, with the conserved charged triad of residues stabilizing CP-CP interactions and the conformation of the sN-IDR kink (Supplementary Fig. 18a, b). We now propose that once RNA binds to the canonical binding site in these species-specific RNA-free CP assemblies, it causes conformational changes in the conserved RNA-binding loop (Fig. 2c, Supplementary Fig. 18a, c). This in turn triggers the movement of adjacent CP[n], CP[n-2] and CP[n-10] units, and the conformational shift of the sN-IDR kink in CP[n] from the 'open' (RNA-free) to the 'closed' (RNA-bound) conformation, along with the reorganization of C-IDR and its binding to RNA at the non-canonical RNA binding site (Fig. 1f, Supplementary Fig. 18c). This model thus combines the interplay between the conserved and species-specific CP-CP and CP-RNA interactions, identified in this study, to propose their role in the assembly of uniform, stable and functional potyviruses.

The assembly of a rod-shape tobacco mosaic virus (TMV; *Tobamovirus tabaci*) occurs spontaneously and does not require external energy sources (e.g., from ATP hydrolysis). It is driven by the laws of thermodynamics, favorable CP-CP and CP-RNA interactions and CP interactions with the aqueous environment[38,39]. The assembly of potyviruses and VLPs is probably also a spontaneous, thermodynamically driven process, as they can readily form in the heterologous system and assemble in the RNA-free VLPs in vitro[14] (Fig. 6b). The lack of ATPase activity in the potyviral CP as a possible energy source also supports this (Fig. 6c).

Earlier studies on potyviruses PVY or pepper vein banding virus (PVBV; *Potyvirus capsivenamaculae*) have suggested that virion assembly proceeds through threading RNA through the central cavity of the disks/rings, causing a conformational shift from the stacked ring to the helical form[40,41]. While the results of our study also support the possibility that the formation of virions or VLP[h+RNA] is preceded by the formation of RNA-free VLPs or at least shorter ordered RNA-free intermediates, the tethering of RNA stem loops through short RNA-free filaments, as observed in TMV[42–44], seems unlikely in the case of potyviruses. Namely, the RNA stem-loop can insert into the lumen of the TMV double ring as the wedge shaped [TMV]CP lacks long IDR extensions[38,45] found in potyviruses[11–13] or potexviruses[46]. In contrast, the presence of disordered long potyviral C-IDRs in RNA-free VLPs (or intermediates) as intermediates would sterically prevent the insertion of the RNA stem loop.

In summary, although potyviruses share a similar overall quaternary structure and folding of the CP unit, the CP-CP and CP-RNA interactions stabilizing the virions are species-specific. Species-specific structural features of CP can influence not only virus assembly and disassembly but also other infection steps involving the CP. During evolution, only those species that manage to establish a balance between variable and highly conserved residues in the CP survive, maintaining the conserved architecture of the virion. We propose that the formation of a species-specific interaction network in virions and RNA-free VLPs (or similar ordered assemblies) enables the formation of uniform and stable virions. The results of this study may aid future research on the molecular mechanisms of the potyvirus infection cycle. Since the structural parameters of potyviral VLPs encapsulating RNA, VLP[h+RNA]s, are very similar to those of virions, they can serve as valid models in biology. The potyviral CP has a high potential for biotechnological or biomedical nanoparticle development, but as we show here, the outcome strongly depends on the amino acid sequence of the CP used.

## Methods
### Preparation and purification of PVA particles
PVA particle preparations were based on an infectious cDNA (icDNA) clone of PVA B11 (Gene bank accession number: AJ296311) expressed under the cauliflower mosaic virus 35S promoter. The PVA icDNA contains the *Renilla* luciferase reporter gene (*rluc*) with a plant intron 1 from ribulose-1,5-bisphosphate carboxylase/oxygenase (Rubisco)[47]. PVA particles carrying the *rluc* containing PVA RNA were purified essentially as in De et al., 2023[48]. Briefly, *N. benthamiana* plants were grown under controlled greenhouse conditions at 22 °C under 16 h (light) and 8 h (dark) photoperiod. PVA infection was initiated at the 4-leaf stage with *Agrobacterium tumefaciens* carrying the PVA infectious complementary DNA construct. *A. tumefaciens* suspension was infiltrated into *N. benthamiana* leaves at a value of optical density at 600 nm of 0.05, following a procedure similar to that described in a previous study[49]. Systemically infected leaves were harvested three weeks post agroinfiltration and stored at −80 °C. The leaves were ground with freshly prepared, chilled Buffer-A (0.1 M sodium phosphate buffer pH 7.4; EDTA, 0.01 M and pH 8; NaOH 0.01 M; β- mercaptoethanol 0.15%; of a volume **V**), using a weight: volume (harvested leaves: buffer) ratio of 1: 2. Leaf debris was removed by centrifugation at 8000 rpm for 20 min in a Thermo Scientific F9-6×1000 Lex rotor and the plant sap was filtered through Miracloth (Merck-Millipore, USA). The sap was mixed with 3% Triton X-100 (3 h), clarified by centrifugation at 9000 rpm for10 min, and the supernatant was mixed with 4% PEG-6000 and 0.2 M NaCl overnight at 4 °C. The PEG precipitate was collected by centrifugation

at 9000 rpm for 20 min and the pellet was dissolved in freshly prepared Buffer-A, using a volume 1/10 of **V**. The dissolved fraction was ultra-centrifuged in a Thermo Scientific fixed angle T-865 rotor at 40000 rpm for 1 h. The pellet was then dissolved in freshly prepared Buffer-B (0.2 M sodium phosphate buffer, pH 7.4; EDTA, 0.01 M; NaOH, 0.01 M; β-mer-captoethanol, 0.15%,) using a volume of 1/20 of **V** and ultracentrifuged through a 30% sucrose cushion in a swing bucket Thermo Scientific AH-629 rotor (23000 rpm, 3 h). The pellet was dissolved in freshly prepared Buffer-A using a volume of 1/200 of **V**. Finally, the dissolved sample was ultra-centrifuged through a 5–40% sucrose gradient in a swing bucket Thermo Scientific TH-641 rotor (22000 rpm, 1 h). Six fractions (2 ml each) and the precipitate were collected and analyzed with SDS-PAGE and western blot using rabbit PVA CP antibodies (SASA, UK) to identify particle distribution. Particle-enriched fractions were then pooled and pelleted to obtain high-quality PVA particles.

### Molecular cloning for bacterial expression of recombinant proteins

The cDNA sequences of [PVA]CP were from PVA isolate B11 (GenBank accession number NC_004039) and PVA isolate Datura (GenBank accession number: Y11426). The cDNA sequence of [PVY]CP originated from the PVY NTN strain (GenBank accession number: KM396648). The cDNA sequence of [TEV]CP was from highly aphid transmissible (HAT) TEV isolate (GenBank accession number: NC_001555), [PepMoV]CP from PepMoV isolate California (GenBank accession number: NC_001517), and [JGMV]CP from JGMV (GenBank accession number: NC_003606). [TEV]CP, [PepMoV]CP and [JGMV]CP sequences were codon-optimized for *E. coli* expression using the VectorBuilder's Codon Optimization tool (https://en.vectorbuilder.com/tool/codon-optimization.html). The gene fragments encoding the CPs were amplified by PCR and individually inserted into the pT7-7 expression vector. The [PVY]CP-MBP construct in the pET28a expression vector was prepared in our previous study[14]. The insert includes the [PVY]CP gene followed by a TEV protease recognition site, then the coding sequences for MBP, an asparagine-rich linker, a factor Xa recognition site, and a hexa-histidine tag. To generate [PVA]CP-MBP, [PVY]CP gene was removed from the modified pET28a vector[14] by PCR amplification of the vector backbone, and a PCR-amplified fragment of the [PVA]CP was inserted in its place. All gene fragments were inserted into their respective vectors using Gibson Assembly[50]. Mutations in [PVA]CP or [PVY]CP genes were generated through site-directed mutagenesis by PCR[51] using pT7-7 plasmids carrying [PVA-Datura]CP or [PVY]CP and mutagenic oligonucleotides (Supplementary Table 3). All recombinant plasmids were transformed into *Escherichia coli* DH5α, purified and their DNA sequences were verified by nucleotide sequencing (Genewiz or Macrogen).

### Expression and purification of VLPs

The expression and purification of all VLPs followed a previously described procedure[14]. Briefly, the pT7-7 vector with the inserted *CP* gene fragment was used to transform *E. coli* BL21[DE3] cells. The cells were grown in 2×YT medium [yeast extract (10 g/l) (BD Biosciences), tryptone (16 g/l) (BD Biosciences), NaCl (5 g/l)] supplemented with 2 mM CaCl₂ and 5 mM MgCl₂ at 37 °C until reaching an optical density at 600 nm between 0.6 and 0.8. Protein expression was induced with 0.1 mM isopropyl-β-d-thiogalactopyranoside overnight at 20 °C. The cell culture was centrifuged at 6,721 rcf for 6 min and the cell pellet was resuspended in phosphate-buffered saline (PBS) [1.8 mM potassium dihydrogen phosphate, 10.1 mM sodium hydrogen phosphate (pH 7.4), 140 mM NaCl, and 2.7 mM KCl]. Cell suspension was sonicated on ice followed by centrifugation (20,000 rcf, 40 min, 4 °C). VLPs were precipitated from the supernatant by incubation with a mixture of 4% PEG 8000 (Sigma) and 0.5 M NaCl for 30 min at 4 °C, followed by centrifugation (16,000 rcf, 30 min, 4 °C). The pellet was resuspended in PBS with 10% (v/v) glycerol (Sigma) by gentle shaking overnight at 4 °C. Insoluble components were removed by centrifugation (35,000 rcf, 30 min, 4 °C). The VLP suspension was loaded onto sucrose density gradient (20 to 60%) and ultracentrifuged (6 h, 117,000 rcf, 4 °C) in a Beckman

50 Ti Rotor. Fractions containing the VLPs were identified by SDS-PAGE analysis. VLPs were dialyzed in PBS, concentrated using Amicon Ultra centrifugal filters (100,000 MWCO, Millipore). For short-term storage, they were kept at 4 °C in PBS, while for long-term storage, they were preserved in PBS with 10% (v/v) glycerol at -70 °C. For the ATPase activity assays, VLPs were further purified, using Heparin Sepharose 6 fast-flow resin (Cytiva), equilibrated at room temperature in 20 mM Tris-HCl (pH 7.4), and subsequently eluted from the column with 100 mM NaCl dissolved in the same buffer. The VLPs samples including this additional purification step were labeled as [PVA]VLP[clean] and [PVY]VLP[clean].

### Expression and purification of other CP constructs

The expression of [PVA]CP-MBP and [PVY]CP-MBP as well as lysis of bacterial cells followed the same procedure as described above for VLPs. *E. coli* BL21[DE3] lysates in PBS were centrifuged for 40 min at 50,000 rcf and 4 °C. The supernatants were filtered through Chromafil Xtra PVDF-20/25 filters with a pore size of 0.2 μm (Macherey-Nagel) and loaded on Ni-nitrilotriacetic acid (Ni-NTA) column (Qiagen) at room temperature. The column was washed with PBS containing 60 mM imidazole. Bound fractions were eluted with PBS containing 300 mM imidazole. Eluted fractions were further purified with gel filtration on a Superdex 200 16/60 PG (GE Healthcare) equilibrated in PBS at room temperature. Purified protein samples were concentrated using Amicon Ultra centrifugal filters (30,000 or 100,000 MWCO, Millipore) and were stored at −70 °C.

### Electrophoresis

SDS-PAGE analysis was performed using NuPAGE 4–12% bis-Tris protein gels (Invitrogen) according to the manufacturer's protocol. The gels were stained with ProBlue Safe Stain (Giotto Biotech).

### Circular dichroism spectroscopy

Circular dichroism (CD) spectra were acquired using a Chirascan CD spectrometer (Applied Photophysics) in the wavelength range of 190 to 250 nm at 20 °C, with a 0.5 nm measuring interval and a dwell time of 1 s, using a 0.1 cm path length cuvette. PVA sample at a concentration of 0.1 mg/ml was prepared in 10 mM sodium phosphate buffer (pH 7.4) (prepared by mixing solutions of 10 mM sodium hydrogen phosphate and 10 mM sodium dihydrogen phosphate until reaching pH 7.4). Each spectrum was averaged over ten consecutive measurements of the same sample.

### Thermal stability assay

Differential scanning fluorimetry measurements were conducted using the LightCycler® 480 System (Roche). PVA at a concentration of 0.05 mg/ml was tested in 50 mM SPG buffers (pH 7.0) containing succinic acid, sodium dihydrogen phosphate, and glycine in a molar ratio of 2:2:7, at different sodium chloride concentrations ranging from 0 to 1 M. The fluorescent dye Sypro Orange (Thermo Fisher Scientific) was added to the mixtures in a 10× final concentration. Continuous fluorescence intensity measurements (excitation wavelength: 470 nm, emission wavelength: 570 nm) were conducted over a temperature gradient from 25 to 95 °C with 3 °C/min increments. Melting temperatures were determined by fitting the Boltzmann function to the raw curves using Origin software version 8.1 (Origin Lab Corporation). The reported melting temperature values (Tm) values represent averages obtained from two independent experiments, each performed in triplicates for each condition (all data available in the Supplementary Data 1 file).

### Molecular cloning for infectious viral cDNA (PVA[CP-R163A])

The MluI-AfeI fragment sequence from the PVA isolate B11 infectious complementary cDNA (icDNA) containing the R163A mutation in the CP sequence was obtained as a synthetic construct from TwistBioscience (USA). The fragment was cloned into an intermediate vector carrying the MluI-AgeI fragment from the PVA icDNA. The MluI-AgeI fragment of the full-length PVA icDNA was then replaced by the fragment carrying the

R163A mutation, and the expression cassette was transferred to the pBIN19 binary vector.

## Dual-luciferase assay

PVA infections were initiated in *N. benthamiana* at the 4-leaf stage with *Agrobacterium tumefaciens* carrying PVA icDNA at optical density 600 nm of 0.05. *Agrobacterium* carrying construct for the expression of firefly luciferase, 35S-fluc-nosT[47] was co-infiltrated at optical density 600 nm of 0.01 to provide an internal control for the normalization of virus-derived *Renilla* reporter activity.

*Renilla* and firefly luciferase reporter activities were measured from leaf samples with the Dual Luciferase Reporter Assay System (Promega, USA) essentially as in Eskelin et al., 2010[47]. For the assay, four leaf disks of diameter 5 mm per plant were pooled, frozen in liquid nitrogen, homogenized, and suspended in 1x passive lysis buffer. Luminescence from *Renilla* and firefly luciferases was measured with a microplate reader (BioTek Synergy H1, Agilent Biotechnologies). Firefly luminescence was used to normalize *Renilla* luciferase activity in samples from agroinfiltrated leaves. Each experiment was repeated three times.

## Immunocapture reverse transcription quantitative polymerase chain reaction (RT-qPCR)

Immunocapture and reverse transcription were performed in tubes pre-coated with anti-PVA CP antibody (1:400 dilution, SASA, UK) for 3 h at 37 °C. Samples consisting of four leaf disks were ground frozen and suspended in extraction buffer (0.13 M NaCl, 1.4 mM $KH_2PO_4$, 8 mM $Na_2HPO_4$, 2.6 mM KCl, 0.05 % Tween20, 2 % Polyvinylpyrrolidone, 0.2 % bovine serum albumin, pH 7.4). The sap was allowed to settle on ice for 1 h before overnight incubation at 4 °C in the antibody-coated tubes. The tubes were washed 3 times with 1x PBS and once with milliQ water prior to cDNA synthesis using random hexamer primers with the RevertAid H minus First Strand cDNA Synthesis kit (Thermo Fisher Scientific, USA). Bound particles were disrupted by 5 min at 65 °C prior to cDNA synthesis according to the manufacturer's instructions. The cDNA was used as a template for the absolute quantitation of vRNA by quantitative PCR with the Maxima SYBR master mix (Thermo Fisher Scientific, USA) using primers (5'-CATGCC-CAGGTATGGTCTTC-3' and 5'-ATCGGAGTGGTTGCAGTGAT-3') targeting the $^{PVA}$CP cistron. A linear standard curve for calculating vRNA copy number was generated with serial dilutions of the plasmid carrying the full-length PVA icDNA expression construct. Each experiment was repeated three times.

## Negative staining transmission electron microscopy (nsTEM)

Purified PVA particles were diluted to a final concentration of 0.1 mg/ml in 10 mM potassium phosphate buffer (pH 8.0) prepared by mixing solutions of 10 mM potassium hydrogen phosphate and 10 mM potassium dihydrogen phosphate until reaching pH 8.0. Various VLPs were diluted to the final concentration of 0.1 mg/ml in PBS. The samples were deposited onto 400-mesh copper grids (SPI Supplies) coated with Formvar film, stabilized with carbon, and glow-discharged using an EM ACE200 (Leica Microsystems). The grids were contrasted with a 1% (w/v) aqueous solution of uranyl acetate. Micrographs were captured using either a CM 100 transmission electron microscope (Philips) at 80 kV, equipped with an Orius SC 200 camera (Gatan) and Digital Micrograph 2.1.1 software, or a TALOS L120 transmission electron microscope (Thermo Fisher Scientific) at 100 kV, equipped with a Ceta 16 M camera and Velox v3.0 software (Thermo Fisher Scientific).

In case of *in planta* analysis of Arg163Ala mutation in the *CP* gene of PVA, PVA particles from infected *N. benthamiana* tissues were visualized by nsTEM at 80 kV with a Jeol JEM-1400 (Jeol Ltd., Japan) instrument equipped with an Orius SC 1000B camera and the Digital Micrograph software version 2.30.542.0 (Gatan Inc. USA). Freshly glow-discharged, carbon-coated, 200-mesh copper grids were coated with anti-PVA CP antibodies (1:100 dilution, SASA, UK) for 1 h at room temperature. Four leaf disks were homogenized frozen, suspended in 0.06 M sodium

phosphate buffer pH 7.4 and incubated for 1 h at 4 °C before an overnight incubation on the antibody-coated grids. The grids were then washed with 10 drops of 0,02 M sodium phosphate buffer pH 7.4 followed by 10 drops of milliQ water and a 5 s stain in 1 % (w/v) neutral uranyl acetate. Excess stain was immediately dried and the grids were stored at RT until electron microscopy. Samples were obtained from three independent experiments.

## N-terminal Edman sequencing

Samples of PVA in 10 mM potassium phosphate buffer (pH 8.0), incubated at 4 °C for about 6 months and $^{PVA}$VLP (PVA Datura CP sequence) in PBS without incubation were analyzed by SDS-PAGE and transferred to PVDF membrane (iBlot 2 PVDF MiniStacks, Invitrogen) using iBlot 2 dry blotting system (Invitrogen). The transfer settings were: 20 V for 1 min, 23 V for 4 min, and 25 V for 2 min. After transfer, the proteins were fixed on the PVDF membrane during a 5-minute incubation in 50% (v/v) ethanol. The membrane was then stained for 1 min in a solution containing 0.1% (w/v) Coomassie R-250 (Sigma), 50% (v/v) ethanol and 5% (v/v) acetic acid, followed by destaining for 5 min in 50% (v/v) ethanol. After air-drying the membrane, the protein bands were excised. They were further destained with 50% (v/v) methanol containing 0.2% (v/v) triethylamine and briefly washed with 100% methanol. N-terminal sequencing of the protein samples from the PVDF membrane was performed using a Protein Sequencer PPSQ-53A Gradient System (Shimadzu, Japan) according to the manufacturer's standard protocol.

## In vitro VLP assembly

The in vitro assembly of $^{PVA}$VLP was performed as previously described[14]. Briefly, $^{PVA}$CP-MBP (in PBS) and TEV protease (prepared in the group) were mixed at a molar ratio of 10:1 (CP-MBP:protease), and DTT was added at a final concentration of 1 mM. The mixture was incubated overnight at 4 °C. The efficiency of cleavage between $^{PVA}$CP and MBP was evaluated with SDS-PAGE and formation of VLPs by cryo-EM.

## ATPase assay

For the ATPase assay[52], the samples of PVA, $^{PVA}$VLP, $^{PVY}$VLP, $^{PVA}$VLP$^{clean}$, $^{PVY}$VLP$^{clean}$, $^{PVA}$CP-MBP, and $^{PVY}$CP-MBP were prepared following the procedures described above. To prepare a control sample, *E. coli* BL21[DE3] cells were transformed with pT7-7 vector devoid of $^{PVA}$CP or $^{PVY}$CP insert and the bacterial lysate was purified using the same VLP purification procedure (described in the »Expression and purification of virus-like particles« section). The bovine serum albumin (BSA) (Zellx Biochem) was used as the negative control and as a positive control, we used the ATPase p97 (Uniprot ID: P55072). The p97 was cloned into the pET24d vector with a hexa-histidine tag and a TEV protease cleavage site at the N-terminus. *E. coli* Rosetta 2(DE3) cells were used for p97 expression. The p97 was isolated using a Ni-NTA column (Qiagen), the hexa-histidine tag was removed using TEV protease, and the protein was further purified using ion exchange chromatography (Resource Q, Cytiva), concentrated, aliquoted, and stored at −80 °C. The ATPase assay was conducted as previously described in ref. 53. Briefly, prior to the ATPase assay, all protein samples were dialyzed in a reaction buffer composed of 50 mM Tris-HCl, 50 mM KCl, and 10 mM $MgCl_2$ (pH 8.0). ATP hydrolysis reaction was initiated by adding 2 mM phosphoenolpyruvate (PEP) (Sigma), 0.3 mM nicotinamide adenine dinucleotide (reduced form; NADH) (Merck Millipore), 5 mM dithiothreitol (DTT) (Sigma), 3 mM adenosine 5'-triphosphate (ATP) (Sigma), and 0.02 U/μl of both pyruvate kinase (Sigma) and lactate dehydrogenase (Sigma) to the reaction mixture. The final concentration of the assayed proteins (PVA, VLPs, CP-MBP, BSA) in all reaction mixtures was 12 μM, except for p97, which was 0.5 μM. Measurements were performed at 37 °C using a BioTek synergy H1 microplate reader with Gen5 software version 3.11 at. ATP hydrolysis was monitored through detection of NADH depletion[52] by continuous measurement of absorbance at 340 nm over two hours (using 14–53 s measuring intervals). Data curves were normalized (Eq. 1) and presented as a change in NADH fraction over time. Each

experiment was repeated three times (Supplementary Fig. 17b).

$$NADH\ fraction = \frac{A - A_{baseline}}{A_{max} - A_{baseline}} \qquad (1)$$

A…Absorbace at 340 nm$A_{baseline}$…Baseline absorbance at 340 nm (measured after complete ATP hydrolysis in the p97 positive control)$A_{max}$…Maximum absorbance at 340 nm (measured at the start of the measurements)

## Mass spectrometry

$^{PVA}$VLP sample was analyzed with SDS-PAGE. Protein bands were excised from the gel and destained in a solution containing 25 mM $NH_4HCO_3$ (Fluka, Seelze, Germany) and 50% (v/v) acetonitrile (Sigma-Aldrich, St. Luis, MO, USA). The solution was discarded, and the gel pieces were dehydrated with 100% (v/v) acetonitrile and dried using a miVac Duo Concentrator (SP Scientific). Proteins were then reduced, alkylated and hydrolyzed with trypsin as described before[54]. One-step reduction and alkylation were carried out by incubating the gel pieces in 25 mM $NH_4HCO_3$ containing 10 mM tris(2-carboxyethyl)phosphine and 40 mM chloroacetamide for 30 min in the dark at room temperature. After washing with 25 mM $NH_4HCO_3$, the gel pieces were dehydrated with 100% (v/v) acetonitrile and dried under vacuum. For in-gel digestion, the proteins were incubated overnight at 37 °C with MS grade modified trypsin (Sigma-Aldrich) at 12.5 ng/µl in 25 mM $NH_4HCO_3$. Peptides were extracted using 50% (v/v) acetonitrile and 5% (v/v) trifluoroacetic acid, concentrated to 10 µl under vacuum, and desalted using in-house prepared StageTips. Tryptic peptides were analyzed using a Compact ESI-qTOF mass spectrometer (Bruker Daltonics, Bremen, Germany) as in a previous study[55]. The obtained spectra were exported as Mascot generic files and analyzed against the *E. coli* taxonomy (23,228 proteins) of the SwissProt database using Mascot software v2.8.3 (Matrix Science Ltd., London, UK). The following parameters were used: 20 ppm peptide and 0.6 Da fragment mass error tolerance, 2 enzyme missed cleavages, carbamidomethyl-Cys as fixed modification, oxidized Met and deamidated Asn and Gln as variable modifications, an automated Decoy database search, 2 unique peptides per protein and 1% target false discovery rate (FDR).

## Cryo-EM sample preparation and data acquisition

PVA in 10 mM potassium phosphate buffer (pH 8.0) (prepared by mixing solutions of 10 mM potassium hydrogen phosphate and 10 mM potassium dihydrogen phosphate until reaching pH 8.0) and various VLPs in PBS, each at a concentration of 1 mg/ml and a volume of 3 µl, were applied onto glow-discharged (GloQube Plus, Quorum) Quantifoil R 2/2 holey carbon grids (Quantifoil). The grids were blotted with the Vitrobot Mark IV (Thermo Fisher Scientific) using a blot force 3 and a blot time 5 s, within an environment of 95% humidity at 4 °C. Datasets were collected on cryo-EM microscope Glacios (Thermo Fisher Scientific) operating at 200 kV and equipped with the Falcon 3EC direct electron detector (Thermo Fisher Scientific). Images were acquired in counting mode with the pixel size of 0.95 Å using the EPU software (Thermo Fisher Scientific). The micrographs were dose-fractionated into 38 to 42 frames with a total dose of 40 e$^-$/Å$^2$. The defocus range used for image collection ranged from −2 µm to −0.8 µm.

## Cryo-EM data processing and model building

All steps of cryo-EM data processing were carried out using cryoSPARC v4.3[56]. The processing steps adhered to the single-particle analysis with helical reconstruction. A general workflow overview is outlined for $^{PVA-B11}$VLP (Supplementary Fig. 19) and $^{JGMV}$VLP (Supplementary Fig. 20) example datasets. Cryo-EM data processing for all other datasets in this study followed the same protocol. FSC curves and cryo-EM maps color-coded according to local resolution of all maps are shown in Supplementary Figs. 19–21. Initial helical parameters estimations for structures with new architectures were obtained using Helixplorer-1 (https://rico.ibs.fr/helixplorer/) (Supplementary Figs. 19, 20, 22, Supplementary Data 2) and

for structures highly similar to PVY or $^{PVY}$VLPs, the initial parameters were taken from previously determined structures of PVY (EMD-0297)[12] and $^{PVY}$VLPs (VLP$^{h7.5}$ - EMD-17046; VLP$^{r8}$ - EMD-17046)[14] (Supplementary Data 2). Helixplorer-1 identifies multiple helical symmetry parameters based on the power spectrum of a helical structure. Generally, multiple symmetry candidates were refined in cryoSPARC v4.3[56]. The final solution was selected after evaluating several plausible reconstructions.

The initial three-dimensional atomic models of the CP units were modeled using the AlphaFold plugin[57] in ChimeraX v1.8[58]. These initial CP models were fitted into their respective cryo-EM maps using ChimeraX v1.8[58]. In PVA and various VLP$^{h+RNA}$s, a model of a pentanucleotide composed of uracil bases (U1-U5), obtained from the PVY structure (PDB ID: 6HXX)[12], was also fitted into the cryo-EM map. The all-uracil base models were used due to helical averaging. The models of PVA and various VLPs fitted in cryo-EM maps were subjected to several rounds of manual refinement in Coot v0.9.4.1[59] and automated real-space refinement in the Phenix 1.20.1 package[60]. The resulting models were validated using Molprobity[61]. The overview of cryo-EM statistics encompassing data processing and model building is in Table 1. ChimeraX v1.8[58] or PyMOL v2.5.5 (Schrödinger, LLC) were used to prepare images. CP-CP interface areas in PVA and various VLPs were calculated with PDBePISA[62]. The electrostatic surface potential was calculated with PDB2PQR[63] in APBS[64] (a plugin in PyMOL v2.5.5 (Schrödinger, LLC)), using the cutoff values of −5 kTe$^{-1}$ and +5 kTe$^{-1}$ for negative and positive potential, respectively.

## Filament length distribution analysis

Filament lengths of VLPs were measured manually from nsTEM micrographs using ImageJ 1.53a software[65]. The filament length distributions were represented as violin plots created in GraphPad Prism v10.2 (GraphPad Software, Inc.), using medium smoothing. The medians, calculated by the software, are represented by dashed black lines on the violin plots. The values of 25th and 75th percentiles, also calculated by the software, are indicated with dashed lines in the same color as their respective plot.

## Amino acid sequence alignment, phylogenetic tree construction, and sequence logos generation

The amino acid sequences of CPs used in the alignments shown in Fig. 3d, Supplementary Figs. 3c, 5b, 11 correspond to those derived from the three-dimensional structures of potyviruses and VLPs analyzed in this and previous structural studies[11–14,20]. The GenBank accession numbers for sequences used in Supplementary Fig. 3c and 11 are provided in the Supplementary Data 1 file. For phylogenetic analysis (Fig. 5a) and sequence logo generation (Fig. 5b), CP amino acid sequences were selected from a subset of GenBank entries used in a previous potyvirus phylogenetic analysis (https://ictv.global/system/files/inline-images/OPSR.Pot_.Fig3_.v15.png)[26], except for the $^{PVY}$CP sequence. The $^{PVY}$CP sequence from this phylogenetic tree differed from the one used here (NTN strain, GenBank accession number: KM396648), so we used the $^{PVY}$CP NTN sequence used in this study. The GenBank accession numbers of sequences used are provided in the Supplementary Data 1 file. CP amino acid sequences used for percent identity matrix and sequence logos in Supplementary Fig. 1 include all *Potyvirus* sequences from the same phylogenetic study[26], except for the $^{PVY}$CP sequence (we used $^{PVY}$CP NTN strain, GenBank accession number: KM396648). GenBank accession numbers used in Supplementary Fig. 1 are provided in the Supplementary Data 1 file. CP amino acid sequences used for sequence logos in Supplementary Fig. 12 were selected from potyviruses of TEV, PVY, and SCMV clades also from the phylogenetic study[26], as well as sequences from different isolates or strains of PVA, TEV, PVY, PepMoV, and JGMV retrieved from the NCBI Nucleotide database. Their GenBank accession numbers are provided in the Supplementary Data 1 file.

All CP amino acid sequence alignments were created using the MUSCLE algorithm[66] available at the European Bioinformatic Institute (EBI) server (https://www.ebi.ac.uk) and were graphically rendered in

ESPript v 3.0[67]. Phylogenetic tree was created using aligned amino acid sequences on the Phylogeny.fr platform (http://www.phylogeny.fr/index.cgi) with the PhyML algorithm[68], which uses the maximum-likelihood method. Nodes with posterior probability values below 0.5 were collapsed. The resulting phylogenetic tree was edited and visualized using FigTree v1.4.4 (http://tree.bio.ed.ac.uk/software/figtree/). All sequence logos were generated with the WebLogo 3 server[69] (https://weblogo.threeplusone.com/) using aligned amino acid sequences.

### Statistics and reproducibility

For statistical analysis of *in planta* experiments (dual-luciferase assays, immunocapture RT-qPCR), data normality was verified with the Kolmogorov-Smirnov and Shapiro-Wilk tests using SPSS Statistics version 29.0.2.0 (IBM, USA) and statistical significance was calculated with Student's t-test[70]. For non-normally distributed data significance was calculated with the Mann-Whitney U-test[71]. A minimum of $n = 3$ replicates was used to derive mean and p-values. Exact sample sizes are provided in the figure legends (Fig. 4e, f).

To ensure reproducibility of the biochemical assays (ATPase assay, thermal stability), all measurements were replicated at least three times. Exact replicate numbers are reported in the Methods sections 'ATPase assay' and 'Thermal stability assay' and corresponding figure legends (Fig. 6c, Supplementary Figs. 4b, 17b).

### Reporting summary

Further information on research design is available in the Nature Portfolio Reporting Summary linked to this article.

### Data availability

Cryo-EM maps and atomic models have been deposited in the Electron Microscopy Data Bank (EMDB) and wwPDB, respectively, with EMDB/PDB accession codes: EMD-53790/9R7R, EMD-53791/9R7S, EMD-53792/9R7T, EMD-53793/9R7U, EMD-53794/9R7V, EMD-53796/9R7X, EMD-53799/9R7Y, EMD-53800/9R7Z, EMD-53801/9R80, EMD-53802/9R81, EMD-53862/9R9W, EMD-53863/9R9X, EMD-53864/9R9Y, EMD-53865/9R9Z, EMD-53866/9RA0, EMD-53867/9RA1, EMD-53868/9RA2, EMD-53632, EMD-53633, EMD-53634, EMD-53635, EMD-53636, EMD-53638, EMD-53639, EMD-53640, EMD-53642, EMD-53643, EMD-53644, EMD-53645, EMD-53646, EMD-53647, EMD-53650, and EMD-53651, with corresponding structures and atomic models provided in Table 1. Raw cryo-EM datasets have been deposited to the Electron Microscopy Public Image Archive (EMPIAR) with accession codes EMPIAR-12819 (EMD-53790/9R7R), EMPIAR-12820 (EMD-53791/9R7S), EMPIAR-12821 (EMD-53792/9R7T, EMD-53793/9R7U), EMPIAR-12822 (EMD-53794/9R7V, EMD-53796/9R7X), EMPIAR-12823 (EMD-53799/9R7Y, EMD-53800/9R7Z, EMD-53801/9R80), EMPIAR-12824 (EMD-53802/9R81), EMPIAR-12825 (EMD-53862/9R9W), EMPIAR-12826 (EMD-53863/9R9X, EMD-53864/9R9Y), EMPIAR-12827 (EMD-53865/9R9Z, EMD-53866/9RA0, EMD-53867/9RA1, EMD-53868/9RA2), EMPIAR-12828 (EMD-53632, EMD-53633, EMD-53634, EMD-53635), EMPIAR-12829 (EMD-53636), EMPIAR-12830 (EMD-53638, EMD-53639, EMD-53640), EMPIAR-12831 (EMD-53642, EMD-53643, EMD-53644, EMD-53645), EMPIAR-12832 (EMD-53646, EMD-53647), EMPIAR-12833 (EMD-53650), and EMPIAR-12834 (EMD-53651). All data are available in the main text, figures, tables, and Supplementary Information file. Source data for all graphs are provided in the Supplementary Data 1 file. Initial and final helical parameters of all filamentous structures from this study are provided in the Supplementary Data 2 file. Uncropped images of gels, supporting Fig. 6e and Supplementary Figs. 2a, 4f, 13c, 13e, 16b and 17a are shown in the Supplementary Information in Supplementary Figs. 23–28, respectively. Other data related to this paper may be requested from the authors.

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

## Acknowledgements

We thank Tanja Peric for technical assistance with cloning and Manca Žagar for technical support with protein purification, and Dr. Matic Kisovec for technical support at the cryo-EM facility. We thank Primož Bembič for providing p97. We thank the Cryo-EM Facility at the Center for Molecular Interactions and Structural Biology of National Institute of Chemistry, which is supported by the Slovenian Research Agency Infrastructure Program IO-0003. We thank Electron Microscopy Unit of the Institute of Biotechnology, University of Helsinki for providing laboratory facilities. The facilities and expertise of the Instruct-HiLIFE Biocomplex unit at the University of Helsinki, a member of Instruct-ERIC Centre Finland, FINStruct, and Biocenter Finland, are gratefully acknowledged. We thank the Slovenian Research Agency for the funding: P1-0391 (MPod AK), J1-4410 (MPod, AK, LK), PhD fellowship (NK), P4-0407 (MTŽ), P1-0207 (AL), as well as the Research Council of Finland grant numbers 1332950 and 1363204 (KM, MPol) and The Ella Georg Ehrnrooth Foundation (SD).

## Author contributions

The study was conceptualized by M. Pod. and N.K. Molecular cloning was conducted by N.K. and L.K. Protein expression and purification was conducted by N.K. and L.K. PVA sample was prepared and *in planta* experiments were performed by S.D., M. Pol., and K.M. Cryo-EM grid preparation, screening, and data collection were conducted by N.K., L.K., and A.K. Data processing, model building, and analysis were done by N.K., as well as biochemical, biophysical and bioinformatic analysis. Negative staining transmission electron microscopy analysis was conducted by M.T.Ž. N-terminal sequencing and mass spectrometry were performed by A.L. The manuscript was prepared by N.K. and M. Pod. All authors provided critical feedback and helped shape the research, analysis and manuscript.

## Competing interests

The authors declare no competing interests.
