## [Transparent Peer Review file · Communications Biology]

Species-specific structural adaptation of the potyviral coat protein in virions and virus-like particles

Corresponding Author: Professor Marjetka Podobnik

This manuscript has been previously reviewed at another journal. This document only contains information relating to versions considered at Communications Biology.

Version 0:

Reviewer comments:

Reviewer #1

(Remarks to the Author)

This is a resubmitted version of a manuscript that I originally reviewed for Nature Communications as Reviewer 1. In my original review I made a number of points that I thought should be addressed before publication. These included the general suitability of the manuscript for publication in Nature Communications.

The new version of the manuscript, intended for publication in Communications Biology, has addressed my previous points as well as those of the other reviewers. I am therefore happy to recommend publication.

Reviewer #3

(Remarks to the Author)

The authors have addressed and clarified all my previous technical concerns. The writing has also improved. I have no further concerns regarding this story.

Point-by-point response to reviewers

Revision of the manuscript sent to Nature Communications, Manuscript No. NCOMMS-25-53179; Koritnik et al., “**Species-specific structural adaptation of the potyviral coat protein in virions and virus-like particles**”. The Editors Dr. Sophie McKenna and Dr. Laura Rodríguez offered the authors of the manuscript the transfer of the revised manuscript to Communications Biology.

Reviewers' comments are in black and authors' answers in blue:

Reviewer #1 (Remarks to the Author):

This manuscript describes a detailed structural analysis, by cryo-EM, of virus particle of potato virus A (PVA) and the structures formed by the coat proteins of various potyviruses when expressed in E.coli. The authors show that altering the amino acid sequence of particular regions of the coat protein influences the structures of the RNA-free virus-like particles that from from the E.coli-expressed material. They also report that the ATPase activity previously reported as being associated with the coat proteins of potyviruses most probably arises from contamination of preparations of coat protein with exogenous ATPase. Finally, the suggest that the different structures adopted by the coat proteins of different potyviruses may have evolutionary significance.

Overall, I found this to be a very detailed and painstaking piece of research. I have no doubt that the results presented are scientifically valid and make a contribution to our understanding of potyviral structures. My main concern is whether the results presented will be of sufficient interest to an audience wider than those who are specifically interested in potyviral structures to justify publication in Nature Communications. In other words, whether the manuscript might be more suitable for publication in a more specialist journal.

We thank the Reviewer 1 for this comment. Potyviruses can infect a wide range of plants and cause severe diseases, resulting in significant economic losses. Therefore, we believe they are, or should be, of general interest, as are studies of structure–function relationships to understand their biology and develop effective control measures. We have added further information on the importance of these viruses in the Introduction, lines 65-78 in the revised manuscript (pdf format).

In addition to this essentially editorial, I have the following more scientific ones:

1. I am not totally convinced by the argument that the different architectures of the VLPs is of evolutionary significance. Obviously, evolutionary trees, such as that presented in Fig. 5a, are based on sequence comparisons. Thus the more distant two viruses are on the tree, the more sequence differences they will have and therefore the more likely they are to have structural differences. Thus the argument about evolutionary significance seems somewhat circular to me. Do the authors have any comments on this?

Answer 1A:

We thank the reviewer for this comment. However, we do not claim that the different architectures of the VLPs are of evolutionary significance. Instead, we suggest that the conserved quaternary structure of the virion influences the evolution (or diversification) of the coat protein (CP) in potyviruses. We

show that three regions (i-iii; Fig. 1d) in the CP, responsible for CP-CP and CP-RNA interactions leading to this structure, have species-specific sequences. This results in differences in the number and chemical nature of CP-CP and CP-RNA interactions between potyviruses. We then suggest that, as long as these different interaction networks preserve the same quaternary structure of the virion, potential mutations in these CP regions are allowed (= evolutionary significance). As we state at the end of the Discussion: During evolution, only those species that manage to establish a balance between variable and highly conserved residues in the CP survive, maintaining the conserved architecture of the virion, lines 774-776.

Indeed, one of the key messages of this study is that the architectures of VLPs formed by potyviral CPs are species-specific. Rather than connecting these different architectures to evolution, we aimed to determine whether there are specific determinants in the CP amino acid sequence causing these differences. To address this, we performed comprehensive structural and mutational analyses, which identified residues in potyviral CPs responsible for these structural differences. We showed that these residues form CP-CP interaction sites (S1-S3), and the fourth, S4, located in the canonical RNA binding site, which affects affinity for RNA as well as intramolecular interactions in CP. We find greater similarity within clades at the S1-S4 sites than between clades (at both sequence and structural levels), as viruses belonging to the same phylogenetic clades have evolved along similar paths. We also show that the identity of the residues at these sites is important. While the amino acid sequences of S1-S4 vary between potyviral CPs, not all changes are tolerated. For example, the mutation of Arg163 (as in PVA-CP) to Ala (as in PVY-CP) is not tolerated in PVA-CP, as it is crucial for RNA-binding and successful viral infection (i.e., evolutionary significance).

2. At several points, the authors discuss the the specificity of interaction between the viral coat protein and its cognate RNA. They seem wedded to the idea that there must be defined sequence on the RNA to enable this. However, Gallo et al. (2018; cited as reference 22 in the text) showed a clear link between replication and assembly which would occur at the same site within the cell. Thus the only RNA the coat protein is exposed to is the replicating viral RNA, obviating the need for recognition sequences. This probably applies to other RNA viruses as well (see, for example, the review by Peyret et al (2025; PMID 39893746). This possibility should be discussed.

Answer 1B:

It seems there has been a bit of a misunderstanding. The Reviewer refers to the specificity in selecting which RNA to encapsidate. In this study, we address the specificity of amino acids involved in CP-RNA interactions within particle structures, which depend on the amino acid sequence of the CP, as demonstrated here with experimental evidence. It has long been known that potyviral CPs bind RNA in a nucleotide sequence non-specific manner. The first article showing this is Merits et al., 1998 (J Gen Virol 79:3123-7. doi: 10.1099/0022-1317-79-12-3123).

We, however, thank the reviewer for this comment. We have now included the suggested reference (Peyret et al, 2025) in the introduction, please see the lines 72-75, to bring up that the major determinant for the selective packaging of viral RNAs is likely the temporal and spatial coupling of replicated RNA and virion assembly. The first article showing this in PVA is Merits et al., 1998 (J Gen Virol 79:3123-7. doi: 10.1099/0022-1317-79-12-3123).

Additionally, in response to Reviewer 1's comment, we have removed two paragraphs discussing assembly initiation in TMV and potyviruses that were previously included in the Discussion (in the revised manuscript, track-changes, text highlighted in blue). We made this change for clarity and to keep focus with the main topic of the manuscript.

Reviewer #2 (Remarks to the Author):

This manuscript describes the structural characterisation virions produced by potato virus A(PVA) – a member of the potyvirus family and comparison to previously published potato virus Y (PVY) virions and a selection of virus-like particles.

These viruses assemble flexible filamentous particles encapsidating their RNA genome in multiple copies of a single capsid protein CP, with N- and C-terminal arms playing a role in long-range interactions directing helical geometry, and in RNA binding.

The Podobnik group have previously published the cryo-EM structure of potato virus Y – and that paper is well cited (81 citations since 2019 – Google Scholar), supporting the impact of structural characterisation of these economically important plant pathogens.

The PVA CP structure is found to be very similar to that of PVY, having a similar pattern of long-range interactions at both N- and C-termini. PVA is found to have some differences in the path of these arms, the authors suggest that there are reduced protein-protein interactions but enhanced protein-RNA interactions – and that these changes are compensatory leading to a net preservation of capsid stability.

Next virus-like particles were assembled by heterologous expression of CP protein in e.coli – leading to VLPS both with- and without- RNA. The varying helical geometries are accommodated by variations on the terminal arms of CP.

Answer 2A:

We thank the Reviewer 2 for this comment. To clarify, we would like to add, that in this paper we show that while the intrinsic structural plasticity of the N-terminal extension (arm) enables varying helical geometries (and the polymorphism within VLPs of one potyvirus species), we also demonstrate the important role of residues from the *globular CP core region* (Figs. 3 and 4) in determining the species-specific architecture of the VLPs, especially the RNA free VLPs (sites S1-S4). This is one of the central discoveries of this study.

I find this paper a challenging read as although the protein-protein interactions are described in great deal and differences between different species and VLP conformations are highlighted, their importance is framed in the context of virus evolution, but the mechanistic explanation of the significance of variation in virion morphology between species is not made clear.

Different patterns of interactions between N-terminal arms that lace capsomeres together in icosahedral plant viruses are well documented, but such differences are interesting phenomena of no clear importance. The argument that these are of mechanistic and evolutionary importance was not clear to me.

The plasticity of CP-CP interactions highlighted by the VLP experiments also seems to argue against the significance of the variations seen between different viral species – there are many ways to form a helix from these proteins. The VLP experiments highlight phenomena of no clear importance. In the authentic virion assembly with a viral genome leads to single conformations, but there are variations between species that could equally be the consequence of a stochastic evolutionary process, having little mechanistic importance.

Answer 2B:

We thank the Reviewer 2 for these comments. Please see the points below for further clarification from our side.

1. The results of this study (which are in line with previous studies, Kežar et al., 2019, Kavčič et al., 2024 and others cited in the manuscript) show that the potyviral coat protein assembles around RNA to form a conserved overall quaternary architecture with conserved left-handed helical symmetry parameters. However, we observe subtle differences between potyviruses in the way their CPs encapsidate RNA, which are due to *three distinct regions of low amino acid conservation* in the extended arms (IDRs) ((i)-(iii), Fig. 1d). These regions influence CP-CP and CP-RNA interactions, and based on our study, this interaction network varies between potyviral species, i.e. is species-specific, in both the number of contacts and the chemical identity of residues that interact. As we demonstrate and discuss, as long as the interaction network enables the formation of the conserved potyvirus structure, mutations in these regions are tolerated (=evolutionary relevance).

2. We also find that the architectures of VLPs (and their degree of structural polymorphism) formed by potyviral CPs are sequence (species-) specific. We find that the residues responsible for this are mainly in the *globular CP core* and form the interaction sites, as identified in this study, S1-S4. These residues influence CP-CP interactions (sites S1-S3) and as well as CP-RNA interactions (S4).

3. We then demonstrate that when the amino acids in sites S1-S4 in ^{PVA}CP are gradually changed to corresponding amino acids from ^{PVY}CP, we induce the formation of stacked-ring VLPs. Vice-versa, we can abolish the formation of the stacked-ring filaments by ^{PVY}CP when selected residues from S1-S4 sites are replaced by those of ^{PVA}CP. Also, as described above in the reply to Reviewer 1, we correctly predicted (structure-based prediction) the crucial role of S4 in RNA binding. In ^{PVA}CP, Arg163 is in the site S4. When mutated to Ala (as found in ^{PVY}CP) it completely loses affinity for RNA binding (Fig. 4d). We then mutated this residue in PVA and demonstrate that such mutation is detrimental for the virus (Fig. 4g) (=evolutionary significance of the site S4). Furthermore, when we replace Ala161 in S4 of ^{PVY}CP with positively charged Arg, then the affinity of CP/VLPs for RNA increases dramatically, resulting in 100% RNA-packing VLPs (Fig. 4h). We are aware that (single point) mutations in proteins such as the potyviral CP can result in stochastic variations in the architecture of nanoparticles composed of CP subunits, as described in our previous study Kavčič et al. 2024, <https://doi.org/10.1038/s42004-024-01100-x>. For the mutations at S1-S4 sites defined in this study, we do recognize that some stochastic structural variation occurs - for instance, the formation of different helical architectures (e.g., h8.7, h9.7, h10.7) in mutant ^{PVA}VLPs (Figs. 4b, c). Nevertheless, it is important to emphasize the structural (e.g., structure-guided design of stacked-ring assemblies) and functional importance (e.g., the role of Arg163 in ^{PVA}CP) of these sites, as shown in this study.

4. Potyviral CPs have a highly conserved triad of charged residues that are important for CP-CP interactions in both, virions (VLP^{h+RNA}) and RNA-free VLPs. If this triad is disturbed, no RNA-free filaments are formed (Fig. 6a, Suppl. Figs. 1b and 18 and Kavčič et al., 2024). The high conservation of these three residues suggests that CPs with mutations at these sites may form less stable virions and/or RNA-free oligomers (intermediates) and such CPs may thus not survive evolutionary pressure. As these three residues are also essential for the formation of RNA-free VLPs, this implies that CPs may form highly ordered CP oligomers during infection cycle (either resembling VLPs shown here or existing as ordered CP oligomers with similar structural parameters). Such RNA-free assemblies may represent intermediates in the formation of structurally uniform and stable virions, as discussed in the Discussion and depicted in *the simple mechanistic model proposed* in Suppl Fig 18.

5. Therefore, our study highlights several regions in potyviral CPs that may affect the viral infection cycle and could aid future studies in the field. As several of these regions are species-specific in terms of amino acid sequence, different potyviruses may respond differently to mutations occurring at equivalent positions in the three-dimensional structure in different ways.

Finally the authors show that previous findings of ATPase activity by CP were the consequence of contamination from the expression system.

Some minor comments:

Line 80 - '.potyvirus genomes are subjected to negative selection with the region encoding CP being most strongly selected' – this seems a very confusing way of saying that CP is highly conserved (or have I misunderstood).

Answer 2C:

Evolution of potyviral CP is under tight negative selection. The diverse functions of CP in the potyvirus infection cycle reinforce the need to eliminate mutations detrimental to the viral viability. Maintaining the structure and stability of the particles necessary for the virus to travel long distances, survive within aphids and infect next plants is an obvious target for negative selection. In the manuscript we also cite a paper Gibbs et al., 2020 (doi:10.3390/v12020132) where the term 'negative selection' (also in context of CP) was used.

Line 101 "The identity of these residues is important because mutations significantly alter the structural parameters of RNA-free VLPs compared to those produced by wild-type CPs, which could represent the filter during evolutionary selection" I do not understand what point is being made here – also it is not clear to me that the mutations that give rise to modest changes in long-range interactions that lead to similarly modest changes in helical geometries are mechanistically important – also the concept of an evolutionary filter is repeatedly used, I do not know what it means and google has not helped me 'evolutionary selection filter' brings up lots of computational data analysis articles.

Answer 2D:

We understand that the term 'evolutionary filter' may be confusing. In the revised text, we no longer use this phrase, but rather rephrase the text, please see close to lines 33-34 in the Abstract, around line 150 in the Introduction, and around the line 667 in the Discussion.

To further address the Answer 2B: in this study, we show that sites S1-S4 influence the architectures of VLPs. We show how architectures of VLPs change when we mutate residues at positions S1-S4. The experiment in which we exchanged residues in ^{PVA}CP with those of ^{PVY}CP, and vice versa, showed that the resulting CP sequences form VLPs that are *significantly* different from VLPs formed by wild type CPs (for example, the ^{PVA}CP mutants mut5 and mut6 form VLPs that are *significantly* different (wider) from VLPs formed by wild type CPs). As these mutated sequences do not exist in nature, we hypothesize that they would most likely result in a defective virus assembly process. This, we believe, is a good starting point for future studies.

Figure 1e – I don't understand the annotation of interactions with CP copies – specifically N+2 in PVY and N-17 in both viruses. N-2 is illustrated in the below panel showing an interaction between K58 and D136 in PVY, the density looks quite poorly resolved and doesn't support accurate side-chain placement.

Answer 2E:

We described interaction network in PVY in details in Kežar et al, 2019 (DOI: 10.1126/sciadv.aaw380, in the main text as well as in Fig 2. and Suppl. Tex S1 and Suppl. Fig. S5); for example, in PVY, if CPⁿ interacts with CPⁿ⁻², then also CPⁿ⁺² interacts with CPⁿ (if one imagines the red extended N-IDR drawn also on the CPⁿ⁻² and so on...). CPⁿ⁺² is not interacting with CPⁿ⁻¹⁷.

Here, the schemes (upper panels for PVA and PVY) compare interactions in PVA with those in PVY (Kežar et al. 2019). The subunits are numbered as $n, n+1, n+2$ until $n+17$, and then the mirror image, $n-1, n-1, \dots, n-17$, please see Kežar et al., 2019 for more details. In these schemes we show that while in PVY, the central CP (CP^n) interacts with 12 other CPs (light orange circles), in PVA the CP^n interacts with only 10 other CPs (light blue), i.e. there are less CP-CP interactions in PVA than in PVY.

The lower two panels show the cryo-EM density of PVA (left; this study) and PVY (right; Kežar et al, 2019, (EMD-0297, PDB id: 6HXX). In PVA, there is no interaction between CP^n and CP^{n-2} (and also mirrored, between CP^{n+2} and CP^n (not shown)). Therefore, the ^{PVA}CP interacts only with 10 CPs and not with 12, as seen in PVY (right lower panel, where we show the interaction between K58 in CP^n and D138 in CP^{n-2} ; the same interaction is between D138 in CP^n and K58 in CP^{n+2} : this is also described in the first chapter of the Results, lines 207-209).

We agree with the Reviewer 2 that the PVY model may not fit ideally into the density, but the orientation of the side chain densities of these two residues (K58 and D138) indicates that they are positioned toward each other, making their interaction very likely. In contrast, interactions are not possible in PVA due to different chemical nature of the side chains and positions of the CP^n/CP^{n-2} interface.

Besides the in Fig. 1E, please see three other angles below:

Line 183 “RNA free helical PVA VLPs have a higher CP stoichiometry and are more compact than those formed by PVY CP” – what is a higher stoichiometry when we are considering a polymer of a single protein?

Answer 2F:

Thank you for this comment. To avoid confusion, we now replaced the word ‘stoichiometry’ with number of CP units per helical turn or per ring, depending on which structure is described.

Reviewer #3 (Remarks to the Author):

In this manuscript, Koritnik et al. describe that while potyviruses share a conserved filamentous virion structure, the specific coat protein (CP) interactions that stabilize this structure are species-specific. The authors demonstrated that this species-specific CP sequence also dictates the architecture and degree of structural polymorphism of RNA-free virus-like particles (VLPs), identifying key amino acid sites that control these features. They validated these findings in planta, showing that a single mutation in an RNA-binding residue prevented virion formation and systemic infection.

I have some concerns regarding the writing, as the manuscript presents an overwhelming amount of structural and biochemical data, most of which are descriptive and challenging for the reader to follow. For example, transition sentences are often lacking, failing to clarify the motivation for determining a particular structure or what the researchers were looking for. Instead, the narrative often reads as if the

experiments were conducted simply because they were possible. Additionally, here are some specific issues that need to be addressed.

Answer 3A:

We thank the reviewer for this comment. We performed this study to address specific research questions (as described in the manuscript and explained also in the answers to Reviewers 1 & 2). In the revised manuscript, we tried to further improve the clarity of the manuscript, the purpose of the study (Introduction, lines 65-78, 135-153), transitions between the chapters (lines 291-296, 319-320, 586-589) and the clarity of Discussion, as marked in the revised text (please see track changes).

(1) I am asked mainly to look at the EM data. In summary, the cryo-EM data and processing are robust, and it's commendable that the raw data were also deposited, as this is not a field requirement. However, a few minor corrections are needed:

a) Box size: Please list the box size in pixels, not Å.

Answer 3B:

Thank you, corrected in the Supplementary Figs. 19 and 20.

b) FSC curve: The FSC curve for PVY-VLP-R208T (h7.5) does not go to zero.

Answer 3C:

We thank the reviewer for this comment. We have corrected the cryo-EM map for PVY-VLP-R208T (h7.5). Now, the FSC curve goes to zero. Earlier, the issue was due to the presence of duplicated particles (25 % of particles were duplicated), which have been removed before re-running the refinement. New cryo-EM map, together with the new half-maps and FSC curve, were deposited to the EMDB under the same ID (EMD- 53643; the old map was replaced). The new cryo-EM map closely resembles the old one. The corrected cryo-EM data can be sent upon request.

Additionally, Fig. 4h, Supplementary Fig. 21 (Supplementary Information, page 23) and Table 1 have been slightly changed - now showing the corrected PVY-VLP-R208T (h7.5) cryo-EM data.

c) Helixplorer: Please clarify whether the Helixplorer software provides a single estimate or if you explored multiple possibilities.

Answer 3D:

Yes, Helixplorer-1 allows exploration of multiple possible helical symmetry parameters that are consistent with the diffraction pattern. We used it to survey geometries compatible with our power spectra and then refined these candidates in cryoSPARC v4.3. The final solution was selected after evaluating several plausible options. We added a sentence about this to the Methods, lines 1070-1073.

(2) In lines 55, 75, and elsewhere, the term "left-helical symmetry" is used. What does that mean? Do you mean a left-handed twist or a left-handed protofilament? Be precise.

Answer 3E:

Thank you for this comment. We mean left-handed helical twist. The term left-handed helical symmetry has been used to describe viruses such as potyviruses and their VLPs in Kežar et al., 2019 and Kavčič et al., 2024. Expressions in the original text such as "left-helical symmetry" were typo. Now we use left-handed helical symmetry throughout the text.

(3) It is not surprising that mutating a few residues in the helical packing interface resulted in a change in packing. I'm curious if helical polymorphism has ever been observed in this type of viruses cultured or purified from nature, rather than from recombinant expressed protein. This would provide valuable context and insight into the biological relevance of the findings.

Answer 3F:

To our knowledge, helical polymorphism has not yet been shown for potyviruses in their natural environment, as they have not been studied as extensively as other viruses.

Otherwise, structural polymorphism has been observed for the rod-shape rigid helical virus barley stripe mosaic virus (BSMV, genus *Hordeivirus*). The wide and narrow form of this virus differ in one CP subunit per helix turn and interactions between the CP subunits (Clare et al., 2015, <https://doi.org/10.1016/j.str.2015.06.028>). The functional difference between the two forms of BSMV virions is not understood, and it has been hypothesized that the wider form represents the intermediate in virion uncoating.

Our work now indicates that structural polymorphism in potyviruses is possible and may be species specific. We hope this study will assist us and other research groups in future studies of the molecular infection mechanisms of these viruses.